# MDAR: A Multi-scene Dynamic Audio Reasoning Benchmark

## Abstract

The ability to reason from audio, including speech, paralinguistic cues, environmental sounds, and music, is essential for AI agents to interact effectively in real-world scenarios. Existing benchmarks mainly focus on static or single-scene settings and do not fully capture scenarios where multiple speakers, unfolding events, and heterogeneous audio sources interact. To address these challenges, we introduce MDAR, a benchmark for evaluating models on complex, multi-scene, and dynamically evolving audio reasoning tasks. MDAR comprises 3,000 carefully curated question–answer pairs linked to diverse audio clips, covering five categories of complex reasoning and spanning three question types. We benchmark 26 state-of-the-art audio language models on MDAR and observe that they exhibit limitations in complex reasoning tasks. On single-choice questions, Qwen2.5-Omni (open-source) achieves 76.67% accuracy, whereas GPT-4o Audio (closed-source) reaches 68.47%; however, GPT-4o Audio substantially outperforms Qwen2.5-Omni on the more challenging multiple-choice and open-ended tasks. Across all three question types, no model achieves 80% performance. These findings underscore the unique challenges posed by MDAR and its value as a benchmark for advancing audio reasoning research. Code and benchmark can be found at https://anonymous.4open.science/r/MDAR-8981.

## 1 Introduction

The ability to reason from audio, including speech, paralinguistic cues, environmental sounds, and music, is essential for AI agents to interact effectively in real-world scenarios. Real-world environments are rich in overlapping auditory signals that convey linguistic content, social cues, temporal patterns, and environmental context (Wei et al., 2022; Nam, 2025). Effective reasoning over these audio streams is critical for AI agents to understand dynamic interactions, anticipate unfolding events, and support decision-making in applications such as autonomous navigation, audio event monitoring, and embodied household agents (Chen et al., 2020; Gao et al., 2023; Yeow et al., 2024).

Existing benchmarks mainly focus on static or single-scene settings and do not fully capture scenarios where multiple speakers, unfolding events, and heterogeneous audio sources interact. Prior work has advanced audio perception and reasoning research, but most existing datasets are restricted to single-scene scenarios, short audio clips, or narrowly scoped tasks (Sakshi et al., 2025; Ma et al., 2025b; Wang et al., 2025b). These limitations make it challenging to evaluate models on complex, temporally evolving audio reasoning, leaving a gap in understanding model capabilities in realistic multi-scene environments.

To address these challenges, we introduce MDAR, a benchmark for evaluating models on challenging, multi-scene, and dynamically evolving audio reasoning tasks. As shown in Figure 1, MDAR comprises 3,000 carefully curated question–answer pairs linked to diverse audio clips, covering five categories of complex reasoning in dynamic audio scenes: **Scene Understanding**, **Social Relationships and Social Rreasoning**, **Event Reasoning**, **Temporal Reasoning**, and **Anomaly Detection and Safety**. Each category targets specific aspects of cognitive reasoning, from inferring social roles and intentions to predicting causal and sequential events. In addition to the common single-choice questions, MDAR also includes open-ended questions that require free-form answers and multiple-choice questions with multiple audios for the first time.

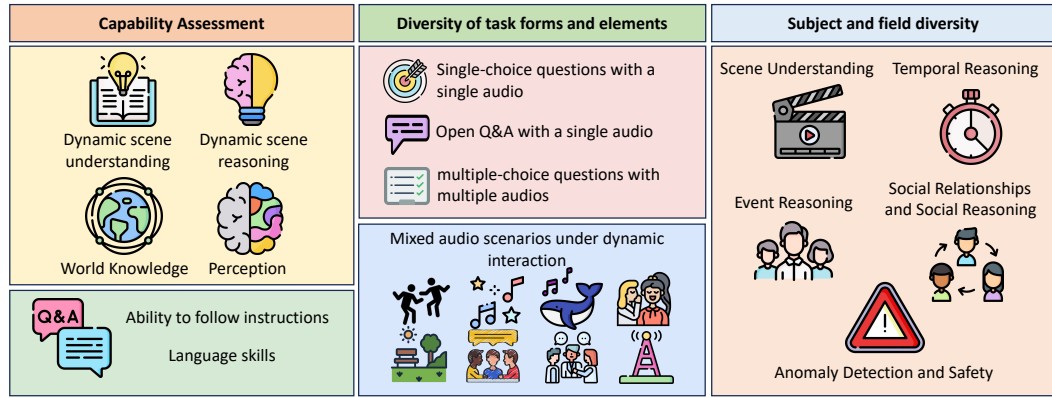

Figure 1: **Overview of MDAR benchmark.** MDAR focuses on five types of complex, multi-scene, and dynamically evolving audio reasoning tasks, spanning three challenging question formats, to test the advanced reasoning, perception, and knowledge capabilities of existing audio models.

We benchmark 26 state-of-the-art audio language models on MDAR and observe that they exhibit limitations in complex reasoning tasks. In the single-choice setting, Qwen2.5-Omni (open-source) achieves 76.67% accuracy, whereas GPT-4o Audio (closed-source) reaches 68.47%. Among different tasks of single-choice questions, Qwen2.5-Omni (Xu et al., 2025a) performs worst on temporal reasoning (71.43%), while GPT-4o Audio struggles most on scene reasoning (61.27%). However, GPT-4o Audio substantially outperforms Qwen2.5-Omni on the more challenging multiple-choice and open-ended tasks. Across all three question types, no model surpasses 80% performance. Overall, these results indicate that even the strongest available models still exhibit significant room for improvement. These findings underscore the unique challenges posed by MDAR and its value as a benchmark for advancing audio reasoning research. Overall, our contributions are three-fold:

1. We introduce MDAR, a large-scale benchmark focusing on evaluating multi-scene and dynamic audio reasoning across five categories, spanning three question types, and for the first time proposing multiple-choice questions with multiple audios.

2. We construct a high-quality data-construction workflow and provide carefully curated audio clips paired with human-annotated questions and answers to systematically evaluate both perceptual and high-level reasoning abilities.

3. We conduct a comprehensive evaluation of state-of-the-art models, revealing the significant challenges posed by MDAR and highlighting key areas for improvement in next-generation audio reasoning agents.

## 2 RELATED WORK

### 2.1 AUDIO-LANGUAGE MODELS

Recent advances in large language models (LLMs) and cross-modal learning have driven substantial progress in audio-language research. Typical approaches jointly leverage speech, music, and general audio data, followed by instruction tuning to endow models with comprehension and instruction-following abilities. Open-source systems such as Qwen-Audio-Chat (Chu et al., 2023), Qwen2-Audio-Instruct (Chu et al., 2024), Audio Flamingo (Ghosh et al., 2025; Goel et al., 2025), SALMONN (Tang et al., 2024), and DeSTA2.5-Audio (Lu et al., 2025) report competitive results. More recent efforts, including Audio-CoT (Ma et al., 2025a), Audio-Reasoner (Xie et al., 2025), and R1-AQA (Li et al., 2025), focus on multi-step reasoning and challenging audio QA. Among proprietary systems, GPT-4o-Audio achieves the highest overall performance, while other commercial models such as Kimi-Audio (KimiTeam et al., 2025) and MiDashengLM (Dinkel et al., 2025) are also evaluated. Fully multimodal models such as Omni-R1 (Zhong et al., 2025) and Qwen2.5-Omni (Xu et al., 2025a) advance unified audio-text reasoning. To provide a holistic comparison, our study additionally benchmarks cascaded audio-language pipelines, enabling a systematic evaluation of state-of-the-art approaches.

Table 1: Comparison between MDAR and other related benchmarks

| | Mixed audio | Multi-audio | Open-ended | Multi-scene within One audio | Chinese | Instruct following |
|---|---|---|---|---|---|---|
| AudioBench | ✗ | ✗ | ✓ | ✗ | ✗ | ✗ |
| AIR-Bench | ✗ | ✗ | ✓ | ✗ | ✗ | ✗ |
| MMAU | ✗ | ✗ | ✓ | ✗ | ✗ | ✗ |
| MMAR | ✓ | ✗ | ✗ | ✗ | ✗ | ✗ |
| **MDAR(Ours)** | ✓ | ✓ | ✓ | ✓ | ✓ | ✓ |

## 2.2 Audio Understanding and Reasoning Benchmarks

Prior works have explored audio question answering and compositional reasoning across speech, music, and environmental sounds. Early efforts include Lipping et al. (2022), a crowdsourced dataset with yes/no and single-word answers, and Ghosh et al. (2024b), which targets order- and attribute-level compositional reasoning with composition-aware fine-tuning for CLAP (Wu et al., 2023). In music, Melechovský et al. (2024) provides a controllable text-to-music system with theory-informed captions, and Weck et al. (2024) offers a human-validated multiple-choice benchmark probing music knowledge. For speech, Zhao et al. (2024) curates large-scale spoken QA in free-form and multiple-choice formats, while Huang et al. (2023) establishes a collaborative instruction-tuning benchmark across diverse speech tasks. Broader evaluations include Wang et al. (2025a), covering speech, scenes, and paralinguistics under instruction following, and Yang et al. (2024), which assesses generative comprehension and chat-based interaction over speech, sounds, and music. More recent reasoning-focused suites such as Sakshi et al. (2025), Ma et al. (2025b), and Wang et al. (2025b) emphasize multi-step perception and domain knowledge, revealing persistent gaps in multimodal integration and deep audio reasoning.

However, these benchmarks mainly target single-scene or static audio scenarios and do not evaluate dynamic, multi-scene audio reasoning, which MDAR aims to address. Our benchmark focuses on evaluating reasoning in dynamic scenarios, including continuous scenarios, causal scenarios, and multi-scenario tasks. We believe this is a key capability requirement for AI agents in real-world scenarios, which current benchmarks do not emphasize. MMAU and MMAR focus on more diverse perceptual samples, including perceptual tasks such as counting and classification. Our evaluation dimension focuses on complex logical reasoning in dynamic scenarios, and is further divided into three tasks: fully mixed audio, multi-audio understanding, and open-ended question answering. Our benchmark examines multiple scenarios within a single audio clip, as well as the model's instruction compliance and Chinese reasoning ability. The comparison is shown in the table 1.

## 3 MDAR BENCHMARK

### 3.1 Overview of Benchmark

**Overview.** MDAR is a benchmark test specifically designed to evaluate the reasoning capabilities of audio-language models in complex and dynamic scenarios. It encompasses 3,000 questions involving high-quality complex reasoning tasks of various types. Figure 2 illustrates examples of different categories and types of questions within this benchmark test. Each example consists of a meticulously designed question and a reference answer. The questions are semi-automatically generated by LLMs and manually annotated by experts and have undergone multiple rounds of screening to ensure their high quality. This benchmark test poses extremely high demands on the complex dynamic reasoning abilities of models, presenting a significant challenge.

**Categories and Subcategories.** Single audio signals or static scenes are insufficient for a comprehensive evaluation of speech agents' reasoning ability. We design three task types: single-choice questions for precise reasoning on complex audio, multiple-choice questions for cross-audio integrative analysis, and open-ended questions for deep reasoning and generation in dynamic scenarios.

**MDAR-main** consists of 1,500 carefully designed single-choice questions, organized into five categories: scene understanding, social relationships and social reasoning, event reasoning, temporal

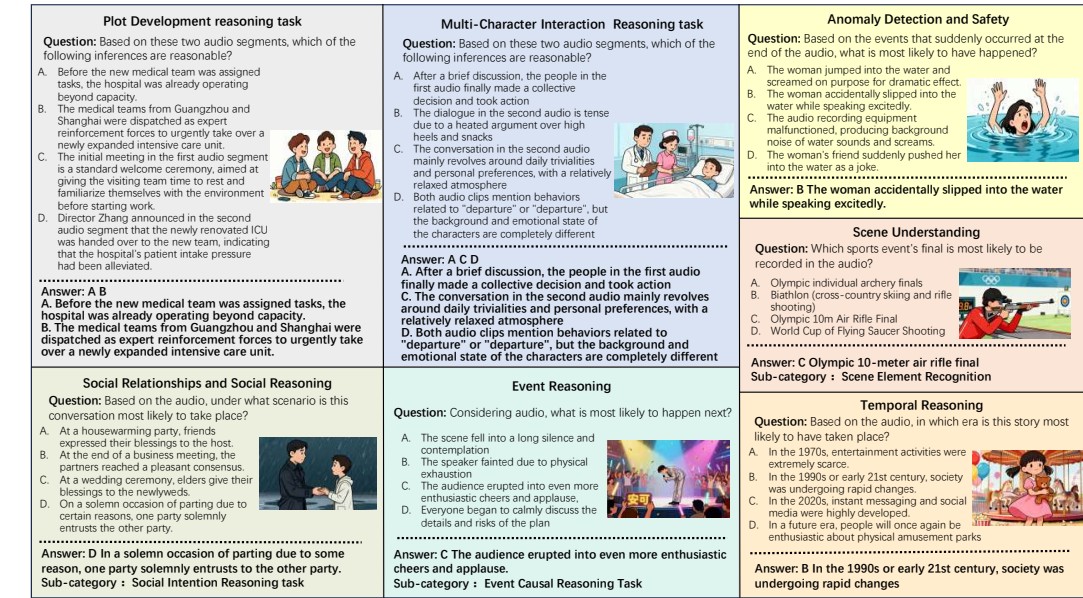

Figure 2: **Examples** from MDAR exquisitely showcase the diversity of the five complex dynamic reasoning tasks and also reveal the design of our newly proposed multiple-choice questions. These tasks not only demonstrate the depth of MDAR in multidimensional capability assessment but also reflect its high standards of challenge.

reasoning, and anomaly detection and safety. Detailed descriptions and evaluation focuses for each category are provided in the Appendix B.

**MDAR-open** anchors its question bank in real-world events and scenarios. The traditional closed-set multiple-choice evaluation method can no longer adequately reflect the core capabilities required by AI agents in real-world open environments. The uncertainty inherent in open-ended questions poses an even greater challenge to the agents' authentic response and judgment abilities. Task categories are the same as MDAR-main.

**MDAR-multi** is the first benchmark to incorporate multi-audio, multiple-choice questions. By embedding real-world events and semantic ambiguities into options, it challenges models to make interpretable decisions even under mishearing, omission, or bias. High-order reasoning tasks are grouped into three dimensions: plot development reasoning, Continuous scene reasoning, and multi-character interaction reasoning, emphasizing knowledge integration and decision-making in real-world contexts, thus providing a finer-grained and more realistic challenge.

**Question Distribution and Difficulty.** Figure 3 shows the distribution of questions across different categories and subcategories and summarizes the key statistics of MDAR. All questions require a comprehensive combination of perception, understanding, and reasoning. The average lengths of questions and answers are 30.68 words and 18.69 words, respectively. The average length of the audio clips is 25.11 seconds,longer than the previous benchmark MMAU which is about 10 seconds and MMAR, which is about 19.94 seconds. Moreover, the three types of questions in MDAR go beyond the single multiple-choice question type in MMAU and MMAR, offering a more complex and comprehensive measure of audio reasoning capabilities.

## 3.2 DATA CONSTRUCTION PIPELINE

As illustrated in the Figure 4, we constructed a high-quality data-construction workflow, which is divided into three main steps. The details of prompts are provided in the Appendix D.

**Data Preparation.** MDAR uses Chinese movies as its metadata source, leveraging multi-threaded narratives and high-production films to construct complex dynamic scenes characterized by openness, high entropy, long temporal dependencies, and strong causality. The metadata set contains 500 films covering a variety of genres and themes (Details are provided in Appendix C). We im-

| Statistics | | Number |
|---|---|---|
| **MDAR-main** | Total Questions | 1500 |
| | Task | 5 |
| | Sub task | 9 |
| | Average question length | 30.68 |
| | Average option length | 18.69 |
| | Average audio length | 25.11 |
| **MDAR-open** | Total Questions | 500 |
| | Task | 5 |
| | Average question length | 41.47 |
| | Average reference length | 175.53 |
| | Average audio length | 25.14 |
| **MDAR-multi** | Total Questions | 1000 |
| | Task | 3 |
| | Average question length | 32.80 |
| | Average answer length | 47.19 |
| | Average audio length | 69.90 |
| | Average answer number | 2.82 |

(a)         (b)

Figure 3: (a) **Distribution** of all types of tasks included in the three benchmark tests illustrates the diversity and complexity of MDAR tasks. (b) **Statistical information** of the three benchmark tests indicates the complexity of the benchmarks through the longer duration of the audio and the rich semantic information.

plemented three sub-processes: **(i) Segment Sampling.** Randomly cropped 20-40-second segments from target films to fit the audio input limitations of existing LALMs while ensuring sufficient length for events and complex information. **(ii) Speaker Diarization.** Using the open-source tool Pyannote to label speaker information in audio, ensuring that multiple speech segments have clear scene events and rich information. **(iii) Global Segment Clustering.** Clustered audio segments with the same speaker to form the audio pairs needed for multiple-choice questions.

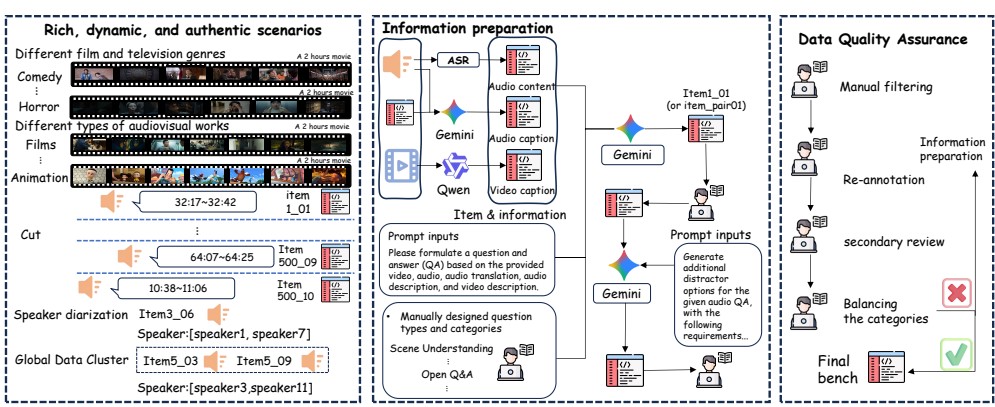

Figure 4: **Data Construction Pipeline**: It consists of three parts: Data Preparation, Audio Pipeline, and Data Quality Assurance, to ensure high quality and consistency of the data.

**Audio Pipeline.** Based on the above corpus, we design a five-step pipeline: **(i) Information Preparation.** We apply FunASR for speech recognition and use Gemini-2.5-pro and Qwen2.5-VL to generate multimodal descriptions, laying the foundation for question construction. **(ii) Framework Construction.** Domain experts are invited to design question types and categories, and to select complex, dynamic scenarios that sufficiently challenge large audio language models (LALMs). **(iii) Content Generation.** Gemini-2.5-pro is used to automatically generate QA pairs, with task-specific prompts tailored for each task type to encourage diversity. For multi-audio QA, special attention is paid to combining independent audio clips with shared contextual information. **(iv) Human-designed screening.** Experts refine and annotate the QA pairs, producing complete metadata including identifiers, timestamps, questions, and answers. Rigorous quality control criteria are applied to remove invalid or irrelevant QA pairs, ensuring that all questions focus on audio content. More screening criteria are provided in the Appendix E. **(v) Distractor generation.** Large language models are used to generate distractor options. Each single-choice question contains four options, while

in multiple-choice questions, distractors are evenly distributed between information from independent and shared audio clips. The final benchmark is output in a standardized JSON format.

**Data Quality Assurance.** We implement multiple traceable steps to ensure high-quality data: **(i) Expert Annotation Filtering.** Manually screening each sample to remove invalid, low-quality, harmful, or irrelevant ones to enhance data purity. Cross-checking by multiple annotators is employed to avoid subjective bias. **(ii) Re-annotation.** Manually correcting original annotation errors and standardizing annotation criteria to improve label accuracy. **(iii) Secondary Review.** Conducting a second screening according to data screening standards to ensure no omissions and no systematic biases. **(iv) Category Balancing.** Ensuring a balanced distribution of question categories to improve the quality of the test benchmark. If category imbalance occurs, returning to the audio process for category completion and iterating until balance is achieved.

### 3.3 EVALUATION METRICS

In this section, we introduce the evaluation methods for MDAR. Since MDAR has different question types, in order to maintain fairness and consistency, we adopt different evaluation methods.

**MDAR-main.** We use accuracy as the evaluation metric. The model is fed with audio, questions, and text instructions to generate answers. Since audio-language models (LALMs) are more adept at producing open-ended responses, we adopt the same response processing pipeline as MMAU. We use robust regular expressions to match the answer strings with the given options and then calculate the accuracy. We found that different prompts can affect the final results. In this evaluation, we used the prompt that yielded the highest average model performance as the final prompt. The results of option distribution and instruction bias are provided in the Appendix I.

**MDAR-open.** We employ a state-of-the-art LLM as an automated evaluator. The evaluation consists of four core components: prompt, question, answer, and reference answer. During the scoring process, the evaluator rates the model's answer on a scale of 0 to 10 based on the given scoring prompt. These prompts consider the usefulness, relevance, accuracy, and comprehensiveness of the answer. Detailed prompt design and scoring examples are provided in the Appendix F.2. We reduce the randomness of scoring by conducting multiple evaluations and swapping the positions of the answers.

**MDAR-multi.** Inspired by SATA-BENCH (Xu et al., 2025b), we use the following four evaluation metrics to measure model performance. For answer set extraction, we still use robust regular expressions to ensure accuracy. **(i) Exact Match (EM)**: For each sample, if the model's predicted answer set is exactly the same as the correct answer set, the accuracy is counted as 1; otherwise, it is counted as 0. The final EM is calculated as the average accuracy. **(ii) Jaccard Index (JI)**: For each sample, we calculate the intersection of the predicted and true answers divided by their union, and then take the average of all samples to measure the degree of overlap. **(iii) Mean Average Precision (Precision)**: For each sample, we calculate the proportion of correctly predicted answers out of all predicted answers, and then take the average of all samples. **(iv) Mean Average Recall (Recall)**: For each sample, we calculate the proportion of correctly predicted answers out of all true answers, and then take the average of all samples. The specific formulas, are provided in the Appendix F.1.

## 4 EXPERIMENTS

### 4.1 SETTING

We evaluated the performance of both cascaded and non-cascaded models on MDAR. The non-cascaded models include large audio-language models, large audio-reasoning models, and full-modal language models. The large audio-language models consist of Qwen-Audio-Chat (Chu et al., 2023), Qwen2-Audio-Instruct (Chu et al., 2024), SALAMONN (Tang et al., 2024), Audio Flamingo 2 (Ghosh et al., 2025), Audio Flamingo 3 Chat (Goel et al., 2025), MiDashengLM (Dinkel et al., 2025), etc. The large audio-reasoning models include Audio-Reasoner (Xie et al., 2025), etc. The full-modal language models include Omni-R1 (Zhong et al., 2025), Qwen2.5-Omni (Xu et al., 2025a), etc. The cascaded models based on captions include GPT-4o Audio+ Qwen2-Audio-Instruct, Qwen2-Audio-Instruct + Llama-3-Instruct, etc. Of course, the models selected for evaluation vary depending on the question type. This is partly because some models lack sufficient

Table 2: Results of models on the MDAR-main benchmark for both audio language models and cascaded models are presented for each category. The best results are highlighted in **bold** and the second-best is underlined. The results cover five task categories: SR (Social Relationships and Social Reasoning), ER (Event Reasoning), SU (Scene Understanding), AD (Anomaly Detection and Safety), and TR (Temporal Reasoning).

| Model | Size | Type | SR | ER | SU | AD | TR | Avg(%) |
|---|---|---|---|---|---|---|---|---|
| Random guess | - | - | 24.40 | 26.06 | 25.04 | 26.38 | 25.41 | 24.78 |
| Human | - | - | 92.64 | 89.24 | 92.13 | 90.69 | 93.14 | 92.66 |
| *Non-cascaded Model* | | | | | | | | |
| Qwen2-Audio-Instruct (Chu et al., 2024) | 7B | LALMs | 38.07 | 31.13 | 23.12 | 35.06 | 27.14 | 33.60 |
| Qwen-Audio-Chat (Chu et al., 2023) | 8.4B | LALMs | 17.93 | 20.34 | 10.98 | 15.52 | 22.86 | 17.73 |
| Audio Flamingo 2 (Ghosh et al., 2025) | 3B | LALMs | 26.37 | 22.54 | 34.29 | 29.41 | 31.61 | 27.73 |
| Audio Flamingo 3 (Goel et al., 2025) | 7B | LALMs | 24.00 | 24.28 | 21.43 | 20.83 | 16.67 | 22.20 |
| Audio Flamingo 3 Chat | 7B | LALMs | 15.26 | 16.18 | 12.86 | 13.73 | 20.69 | 15.47 |
| Kimi-Audio-Instruct (KimiTeam et al., 2025) | 7B | LALMs | 14.52 | 18.38 | 13.29 | 20.69 | 25.71 | 16.67 |
| Omni-R1 (Zhong et al., 2025) | 7B | OLMs | 44.89 | 45.83 | 45.66 | 60.34 | 38.57 | 46.73 |
| R1-AQA (Li et al., 2025) | 7B | LALMs | 39.11 | 41.18 | 27.17 | 35.06 | 38.57 | 37.80 |
| SALAMONN (Tang et al., 2024) | 7B | LALMs | 36.45 | 29.76 | 37.55 | 35.29 | 37.89 | 34.75 |
| Audio-Reasoner (Xie et al., 2025) | 8.4B | LALMs | 45.93 | 42.65 | 43.35 | 37.36 | 47.14 | 43.80 |
| DeSTA2.5-Audio (Lu et al., 2025) | 8B | LALMs | 63.41 | 62.01 | 54.91 | 53.45 | 64.29 | 60.93 |
| MiDashengLM (Dinkel et al., 2025) | 7B | LALMs | 68.44 | 65.69 | 62.43 | **74.71** | 70.00 | 67.80 |
| GPT-4o mini Audio | - | LALMs | 61.19 | 54.34 | 62.86 | 66.67 | 56.90 | 61.47 |
| GPT-4o Audio | - | LALMs | 68.15 | 73.53 | 61.27 | 63.22 | 72.86 | 68.47 |
| Qwen2.5-Omni (Xu et al., 2025a) | 3B | OLMs | 63.26 | 66.67 | 63.58 | 58.62 | 61.43 | 63.60 |
| Qwen2.5-Omni (Xu et al., 2025a) | 7B | OLMs | **78.67** | **75.98** | **75.72** | 73.56 | 71.43 | **76.67** |
| *Cascaded Model* | | | | | | | | |
| GPT-4o Audio + Qwen2.5-Omni | 7B | - | 56.59 | 53.68 | 50.87 | 55.75 | 54.29 | 54.93 |
| GPT-4o Audio + Qwen2-Audio-Instruct | 7B | LALMs | 33.19 | 24.51 | 21.39 | 28.74 | 37.14 | 29.13 |
| GPT-4o Audio + Llama-3-Ins. | 8B | LLMs | 55.70 | 48.77 | 46.82 | 60.92 | 58.57 | 53.53 |
| GPT-4o Audio + DeepSeek-V3 | - | LLMs | **82.52** | **76.88** | **87.14** | **82.84** | **82.18** | **82.13** |
| GPT-4o Audio + DeepSeek-R1 | - | LLMs | 46.52 | 39.31 | 34.29 | 41.42 | 43.10 | 43.33 |
| Qwen2-Audio-Instruct + Llama-3-Instruct | 8B | LLMs | 54.62 | 45.41 | 54.74 | 54.43 | 57.98 | 53.93 |
| Qwen2-Audio-Instruct + GPT-4o Audio | - | LALMs | 59.41 | 49.71 | 68.57 | 60.29 | 65.52 | 59.67 |
| Qwen2-Audio-Instruct + DeepSeek-R1 | - | LLMs | 42.52 | 39.95 | 30.64 | 37.36 | 34.29 | 39.47 |
| Qwen2-Audio-Instruct + Qwen2.5-Omni | 7B | - | 49.04 | 46.57 | 40.46 | 48.28 | 45.71 | 47.13 |
| Qwen2-Audio-Instruct + DeepSeek-V3 | - | LLMs | 76.44 | 74.29 | 76.44 | 77.01 | 67.63 | 74.80 |

Chinese capabilities or do not support Chinese audio understanding, and partly because some existing models do not accept multimodal inputs with multiple voices. Note that since our benchmark is automatically synthesized by Gemini, the results of Gemini will not be included in the comparison. Regarding the selected and unselected models as well as detailed implementation details, please refer to the Appendix G.

## 4.2 RESULTS AND DISCUSSION

Tables 2, 3, and 4 show the results of mainstream models on all tasks of MDAR-main, MDAR-open, and MDAR-multi. Results for the subtasks of MDAR-multi are provided in the Appendix H.

**Our Benchmark is Challenging.** From the three tables, we can observe that the models' performance on the bench is as follows: the best-performing model in the multiple-choice section, Qwen2.5-Omni, achieved only 76.67%. The highest score in open-ended questions was only 7.46, and the highest score among open-source models was only 6.58. In the benchmark, the majority of models scored low, with many models having single-digit accuracy rates. This indicates that MDAR is highly difficult and extremely challenging. Moreover, there is a significant difference in performance among different models. In the multiple-choice section, models such as Audio Flamingo 3 Chat and Kimi-Audio-Instruct did not exceed 20%. The scores in multiple-choice and open-ended questions varied by more than 340%, demonstrating that MDAR has good discriminability.

**The Gap Between Open-Source and Closed-Source Models Remains Significant.** According to Figure 5, we observed that only performance of Qwen2.5-Omni in the multiple-choice benchmark exceeded that of GPT-4o Audio. The performance of other open-source models still could not

Table 3: Scores of models on the MDAR-open benchmark for audio language models are presented for each category. The best results are highlighted in **bold** and the second-best is underlined. The five categories are represented in the same way as in Table 2.

| Model | Size | Type | SR | ER | SU | AD | TR | Avg |
|---|---|---|---|---|---|---|---|---|
| Qwen2-Audio-Instruct | 7B | LALMs | 6.70 | 6.56 | 5.89 | 6.81 | **6.53** | 6.58 |
| Qwen-Audio-Chat | 8.4B | LALMs | 3.24 | 3.77 | 4.30 | 2.44 | 5.50 | 3.58 |
| Audio Flamingo 2 | 3B | LALMs | 1.57 | 1.87 | 2.32 | 2.37 | 1.00 | 1.78 |
| Audio Flamingo 3 | 7B | LALMs | 3.19 | 3.36 | 3.84 | 4.44 | 3.50 | 3.33 |
| Audio Flamingo 3 Chat | 7B | LALMs | 1.58 | 1.81 | 1.91 | 2.82 | 1.00 | 1.73 |
| Qwen2.5-Omni | 3B | OLMs | 3.76 | 3.85 | 4.27 | 2.90 | 2.00 | 3.82 |
| Qwen2.5-Omni | 7B | OLMs | 4.53 | 4.62 | 4.79 | 3.81 | 2.56 | 4.58 |
| GPT-4o Audio | - | LALMs | **7.62** | **7.34** | **7.60** | **7.42** | 5.50 | **7.46** |
| GPT-4o mini Audio | - | LALMs | 5.57 | 5.14 | 4.64 | 6.53 | 5.50 | 5.30 |
| DeSTA2.5-Audio | 8B | LALMs | 3.67 | 3.61 | 3.89 | 3.99 | 1.50 | 3.65 |
| Kimi-Audio-Instruct | 7B | LALMs | 3.98 | 3.56 | 3.92 | 3.06 | 4.00 | 3.75 |
| MiDashengLM | 7B | LALMs | 3.84 | 3.64 | 4.15 | 3.52 | 4.00 | 3.75 |
| Omni-R1 | 7B | OLMs | 1.56 | 1.73 | 1.69 | 2.39 | 1.00 | 1.66 |
| R1-AQA | 7B | LALMs | 4.36 | 4.30 | 4.82 | 5.26 | 3.50 | 4.36 |
| SALAMONN | 7B | LALMs | 2.34 | 2.47 | 2.49 | 2.81 | 2.50 | 2.41 |
| Audio-Reasoner | 8.4B | LALMs | 3.23 | 3.41 | 3.65 | 2.16 | 2.06 | 3.34 |

surpass that of the closed-source model, and there was a significant gap. This phenomenon was even more pronounced in the open-ended question-answering benchmark and the multiple-choice benchmark. Score of GPT-4o Audio in the open-ended question-answering benchmark was 13.37% higher than the best open-source model, and its metrics in the multiple-choice benchmark exceeded those of the best open-source model by 92.78%. This indicates that open-source models still have significant shortcomings in generating answers and handling multiple inputs.

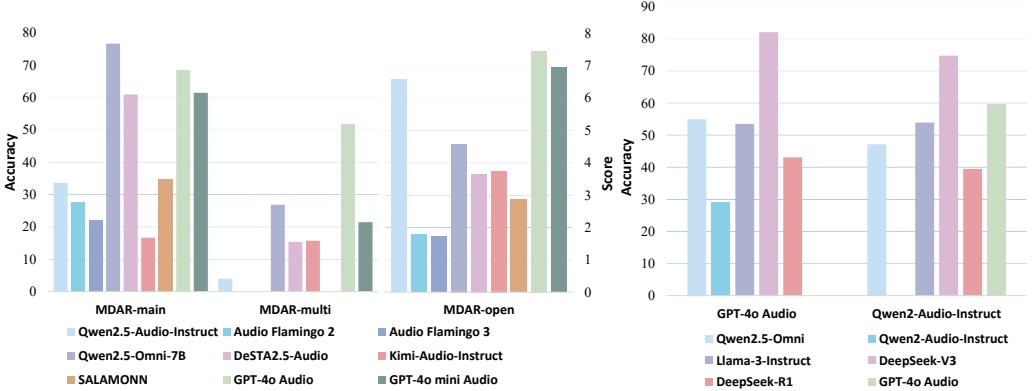

Figure 5: (a) **Left:** The gap between open-source and closed-source models on three testing benchmarks. The closed-source models significantly outperform the open-source models. (b) **Right:** Comparison of different scenarios with the same caption model and the same reasoning model in the cascaded models.

**Analysis of Perception and Reasoning Capabilities of Cascaded caption Models.** As shown in Figure 5, the experiments focused on the single-choice benchmark using GPT-4o Audio (the best closed-source model) and Qwen2-Audio-Instruct (the baseline open-source model). These two models were used as caption models and combined with other models for reasoning. The results in the latter half of Table 2 show that: **(i)** Under the same caption conditions, combinations of GPT-4o Audio with other models performed worse than GPT-4o Audio alone, indicating its strong perception capabilities. **(ii)** Combinations of Qwen2-Audio-Instruct with other models outperformed Qwen2-Audio-Instruct alone, suggesting its weaker perception capabilities. **(iii)** Among the same reasoning models, the combination of GPT-4o Audio and Qwen2.5-Omni outperformed the combination of Qwen2-Audio-Instruct and Qwen2.5-Omni, proving that GPT-4o Audio has stronger perception capabilities. DeepSeek-V3, when used as a reasoning model, achieved results that far exceeded those obtained by combining the same caption model with other reasoning models, and was even comparable to the best non-cascaded models, demonstrating its significant reasoning potential.

Table 4: Results of models on the MDAR-multi benchmark for audio language models are presented. Detailed results for each category can be found in the Appendix H The best results are highlighted in **bold** and the second-best is underlined. The explanations for the four metrics are provided in 3.3.

| Model | Size | Type | EM(%) ↑ | JI(%) ↑ | Precision(%) ↑ | Recall(%) ↑ |
|---|---|---|---|---|---|---|
| Human | - | - | 87.6 | 91.52 | 86.44 | 94.12 |
| Qwen2-Audio-Instruct | 7B | LALMs | 4.00 | 22.48 | 28.22 | 16.67 |
| Qwen-Audio-Chat | 8.4B | LALMs | 6.17 | 42.77 | 56.67 | 54.02 |
| Qwen2.5-Omni | 3B | OLMs | 25.67 | 57.98 | 62.80 | 52.87 |
| Qwen2.5-Omni | 7B | OLMs | 26.88 | 57.97 | 61.85 | 50.00 |
| GPT-4o Audio | - | LALMs | **51.82** | **78.65** | **77.26** | **83.91** |
| GPT-4o mini Audio | - | LALMs | 21.45 | 53.23 | 72.97 | 55.41 |
| DeSTA2.5-Audio | 8B | LALMs | 15.38 | 55.47 | 47.05 | 51.13 |
| Kimi-Audio-Instruct | 7B | LALMs | 15.74 | 69.97 | 53.47 | 53.45 |
| MiDashengLM | 7B | LALMs | 8.72 | 51.60 | 52.60 | 48.96 |
| Omni-R1 | 7B | OLMs | 3.63 | 51.17 | 62.50 | 63.85 |
| R1-AQA | 7B | LALMs | 2.42 | 18.58 | 18.77 | 18.07 |

**Exploring the Limitations of Perception and Reasoning Capabilities.** To explore whether the limitations of current models lie in perception or reasoning, and to identify the capabilities, deficiencies, and potential future improvement directions of mainstream LALMs, we conducted two experiments:

**Gaussian Noise Replacement Experiment:** Taking four different types of models as representatives, we replaced the audio with Gaussian white noise of the same length and input it into the audio-language models. As shown in Figure 6, the accuracy of the models dropped significantly when they received white noise, approaching random guessing. This indicates that the models received audio information. However, the results also show that even with noisy audio, the models were still able to derive some answers from the textual questions.

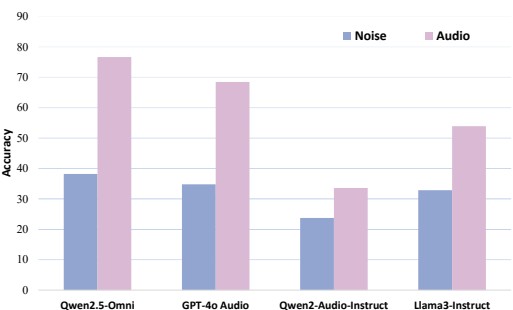

Figure 6: When Gaussian noise was used to replace the audio on MDAR to test GPT-4o Audio, Qwen2-Audio-Instruct, Qwen2.5-Omni, and Qwen2-Audio-Instruct + Qwen2.5-Omni, the performance of all models experienced a significant drop.

**Error Analysis:** We classified the incorrect answers of four models using Gemini-2.5-flash and found that inference errors were the main source of error for most models, such as GPT-4o Audio with a inference error rate of 86.68%. There are various reasons for model errors, and LALMs tend to answer more content, while comparative text QA only needs to answer A/B/C/D. In addition, the problem of hallucinations mainly manifests as creating non-existent sound events or options. Models such as SALMONN miss key content due to insufficient perceptual granularity, resulting in incorrect selection. This suggests that model training can develop towards finer perceptual speech descriptions and fine-grained QA problem training.

In models with parameters exceeding 7B, there are fewer knowledge gap errors, because large-scale pre training already covers basic audio segments. Different models exhibit different preferences in terms of formatting errors, such as Omni-R1 often adding line breaks, and some models can only partially restate options. LALMs have issues with randomness and instability, and different prompts and runs can lead to significant differences in accuracy. This suggests that model training can improve the stability of the model for different format problems through prompts in different formats, especially when dealing with text requirements other than speech. Detailed analysis and case studies are provided in Appendix I and Appendix J.

**The Correlation Analysis between LLM and Human.** we randomly selected 10% of the sample and manually scored the answers of Qwen2-Audio-7B, Qwen25-Omni-7B, and GPT-4o Audio models. The range of manual scoring was 0-5 points, with 5 points indicating complete correctness and clear expression. Then, Pearson was given, with all three coefficients greater than 0.9, indicating

that our human evaluation was highly correlated and reliable with LLM evaluation. At the same time, we conducted paired sample t-tests between all three models based on open benchmarks, and the P-values were all less than 0.05, indicating that there were significant differences in the evaluation scores of different models. Our evaluation was completely effective. The following table 6 and table 5 shows two results.

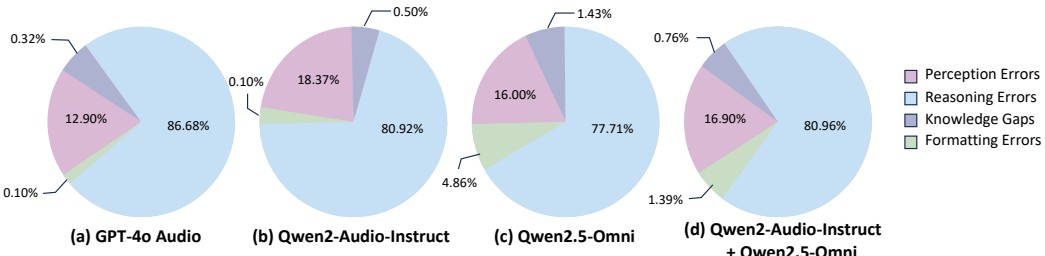

Figure 7: **Distribution of error types** in the responses of four models. Reasoning errors remain the primary type of errors in the responses of LALMs, followed by perception errors.

Table 5: Pearson correlation coefficient between LLM and human ratings

| | Sample | Qwen2-Audio-Instruct | Qwen2.5-Omni | GPT4o-Audio |
|---|---|---|---|---|
| Pearson | 50 | 0.95312 | 0.9268 | 0.9714 |

Table 6: T-test results of the correlation between LLM and human ratings

| | Sample | Qwen2-Audio-Instruct | Qwen2.5-Omni | GPT4o-Audio |
|---|---|---|---|---|
| t | 500 | T: 6.421, P: 0.000 | T: -11.866, P: 0.000 | T: -8.389, P: 0.000 |

**Suggestions to guide the development of LALMs.** Based on the benchmark of this article, we will provide some suggestions to guide the development of LALMs from the results. We divided the model into two stages: with and without RL training. We found that the R1-AQA model, which had undergone RL on speech data, performed better than qwen2Audio-7b, indicating that RL on speech data may be effective. In MDAR testing, the improvement brought by RL is minimal. I speculate that the main reason is that LALMs require a more powerful base model, and I believe that the effect of RL will be more significant at that time. We also compared the training stages and parameter sizes of the model. Under the same architecture, the performance of the 7B model has been improved compared to the 3B model, but as shown in the error analysis, the 7B model mainly analyzed in this paper has the second largest proportion of perception error. To address this issue, I believe we should focus more on the quality of the data (such as fine-grained problems) and better post training processes, rather than continuing to increase the number of parameters. In the cascade experiment, we observed a significant improvement in the accuracy of the inference model using deepseek-V3, which not only proves the insufficient text inference ability of existing LALMs, but also provides us with some potential important data insights. When training LALMs without losing perceptual ability, we should focus on data diversity (to alleviate the instability shown in real-time experiments) and text specific abilities. I believe that adding code or mathematical training can help improve speech reasoning ability.

## 5 CONCLUSION

We introduce MDAR, a benchmark for evaluating models on challenging, multi-scene, dynamically evolving audio reasoning tasks. MDAR comprises 3,000 carefully curated question–answer pairs linked to diverse audio clips, covering five categories of complex reasoning across three question types. We evaluated 26 audio models, revealing that MDAR poses substantial challenges across question formats and evaluation methods. Furthermore, we analyzed gaps between open-source and closed-source models, cascaded caption models' perception and reasoning capabilities, and overall limitations in model audio understanding. These findings underscore the challenges posed by MDAR and its value as a benchmark for advancing audio reasoning research.

## 6 ETHICAL STATEMENT

This study strictly adheres to the ethical norms of academic research. All models and data used are collected in accordance with the principle of fair use, and do not contain identifiable personal information. During the model evaluation process, no direct harm was inflicted on any specific group, nor was any discriminatory or offensive content introduced.

## 7 REPRODUCIBILITY STATEMENT

In order to ensure the reproducibility of the research, we have detailed the data sources, model information, hyperparameter configurations, evaluation strategies, and evaluation metrics in the paper. All experiments were conducted in the specified hardware and software environment. We open source the benchmark and evaluation scripts, and the complete data will be released upon acceptance of the paper, ensuring that other researchers can replicate the experimental results under the same conditions and further carry out research based on this.

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

## A USAGE OF LLM

During the process of writing the paper, we only used the Large Language Model (LLM) to polish some of the text in order to enhance the fluency and accuracy of the language expression. All research content, viewpoints, and data are original. The LLM was used only as an auxiliary tool and did not participate in the creation of the core content.

## B TASK

### B.1 TASK INTRODUCTION

We divide the benchmark tests into five major task categories based on the idea of enhancing the complex dynamic audio capabilities and real mixed-scene reasoning capabilities of audio-language models. Below are detailed introductions to each category and the focus of the benchmark assessments.

**Scene Understanding** focuses on the holistic perception and semantic interpretation of complex, dynamic, multimodal scenarios. It is divided into three subtasks: Scene Localization, Scene Change Reasoning, and Scene Element Recognition. This category tests the models' ability to extract key entities, attributes, and states from multimodal information sources such as text and audio. It also evaluates their capacity to distinguish between common-sense and counter-common-sense elements in open-world scenarios, as well as to understand spatial layouts, functional relationships between objects, and causal constraints.

**Social Relationships and Social Reasoning** centers on modeling implicit social relationships, role identities, emotional motivations, and normative constraints between individuals. It consists of two subtasks: Social Intent Reasoning and Character Identity Association. This category assesses the models' ability to infer intimacy, power distance, and trust levels from dialogues, micro-expressions,

gestures, and historical interactions. It also examines their capability to reason about the facade and subtext in multi-party interactions—such as irony, politeness, and implicit requests.

**Event Reasoning** emphasizes modeling and counterfactual thinking about the causal, conditional, and intervention effects of multi-step event chains. It is divided into two subtasks: Event Causal Reasoning and Event Sequence Reasoning. This category tests the models' ability to handle causal graphs with multiple causes for one effect and one cause for multiple effects.

**Temporal Reasoning** focuses on precisely modeling explicit and implicit temporal information, temporal constraints, and cross-scale dynamic evolution. It evaluates the models' ability to provide accurate answers to vague temporal questions and to infer the temporal priority relationships between events.

**Anomaly Detection and Safety** centers on rapidly identifying audio that deviates from the norm or implies risks in an open environment and proposing interpretable safety intervention strategies. This category tests the models' ability to locate subtle anomalies in multimodal inputs and to generate corresponding emergency response plans.

**Plot Development Reasoning** focuses on the ability to model the coherence of narrative logic, the evolution path of conflicts, and the possibility of outcomes. It tests the model's ability to infer reasonable directions for subsequent plot development, key turning points, and analyze the impact of different choices on the final narrative result in multi-scenario contexts, based on current plot clues, character personality settings, and potential contradictions.

**Continuous Scene Reasoning** focuses on the ability to model the spatial correlation between scenes, changes in environmental states, and the logic of information transmission. It tests the model's ability to infer the relationships between different scenes, integrate key information scattered across multiple scenes, and analyze the impact of scene changes on character behaviors or event progression, from continuously switching scenes.

**Multi-Character Interaction Reasoning** focuses on the ability to model goal conflicts, interest correlations, and interaction strategies among multiple subjects. It tests the model's ability to infer the core goals and potential demands of each character, analyze the cooperative or conflicting interest relationships between characters, and interpret the underlying motivations behind complex interactive behaviors, based on the dialogue content, behavioral performances, and historical interaction records of multiple characters.

## B.2 TASK DIFFICULTY

Based on the experimental data in Table 2, Table 3, Table 4, and Table 8, it can be found that: among the core task types of single-choice questions and open-ended questions, the scene understanding task is the most difficult, while the anomaly detection and event reasoning tasks also present certain challenges. Further analysis by question type shows that: in the context of open-ended questions, temporal reasoning is the most difficult task type, followed by the anomaly detection task in terms of difficulty; in the context of multiple-choice questions, the difficulty coefficient of the continuous scene reasoning task reaches the highest level. This result indicates that the current model still has significant limitations in its reasoning ability when dealing with dynamically switching continuous scenes, which is a key direction that requires focused breakthroughs in subsequent optimizations.

## C DATA SOURCE

The MDAR Metadata Movie Collection includes 500 carefully selected Chinese films, covering a wide range of genres and themes. It is designed to construct complex and dynamic scenarios, so as to meet the requirements of multi-threaded narratives and high production standards. Below is a detailed statistical breakdown of the collection. The 500 films in the MDAR Metadata Movie Collection are categorized by common film genres and themes, with the statistical data presented in the Table 7.

The MDAR Metadata Movie Collection has the following characteristics:

Table 7: Data Statistics of metadata

|  | Types | Number |
|---|---|---|
| | Horror | 10 |
| | Comedy | 73 |
| | Martial Arts/Action | 48 |
| | Animation | 44 |
| 11 types of films | Romance | 94 |
| | Sci-Fi | 12 |
| | War | 25 |
| | Suspense/Crime | 88 |
| | Inspirational Drama | 62 |
| | Period Drama | 14 |
| | Other | 30 |
| | Total Films | 500 |

- **Openness:** The film content is rich and diverse, covering various genres and themes, which provides a broad perspective for research.
- **High Entropy:** The film plots are complex, involving multiple possibilities and uncertainties, which increases the complexity of the data.
- **Long Temporal Sequence:** Films typically span a long time period, enabling the portrayal of character growth and event development.
- **Strong Causality:** There are clear causal relationships between plots and events in the films, which facilitates the analysis and understanding of complex dynamic scenarios.

This data provides a solid foundation for the construction of the MDAR benchmark and also offers abundant resources for relevant research.

## D PROMPT OF DATA CONSTRUCTION PIPELINE

---

**Prompt of Data Generation in MDAR-main**

**System:**
You are a helpful AI assistant, designed to provide useful QA pair to customers. The user will provide you with a video, audio, audio translation, audio description, and video description. Please refer to this desired output and provide your own unique response.

**User:** Based on the provided video, audio, audio translation, audio description, and video description, create three questions about the audio. These should be three audio reasoning questions with options and answers.

The audio translation is [**audio translation**], the audio description is [**audio description**], and the video description is [**video description**].

The questions must meet **the following criteria**:

1.They should be reasoning questions related to the audio.

2.The category of the questions should fall into one of the following: temporal reasoning task, social intent reasoning task, anomaly detection and safety task, and specify which category it belongs to.

3.The posed questions should include a thought chain explanation.

---

4.The questions cannot be answered solely by the text of the title, they must require reasoning based on both the question and the audio.

5.The questions should introduce some additional reasoning information that does not exist in the audio to increase difficulty.

6.Various information from the audio should not appear directly in the question text; just pose the question.

7.The answers provided should follow this format: Question:..... Question Category:..... Question Options:...... Question Answer:......

**Assistant:**
{response}

---

Prompt of Data generation in MDAR-multi

**System:**
You are a helpful AI assistant, designed to provide useful answer to customers. The user will provide you with a video, audio, audio translation, audio description, and video description. Please refer to this desired output and provide your own unique response.

**User:** Based on the two provided relevant audio segments along with their respective translations, descriptions, and video descriptions, formulate multiple-choice audio reasoning questions-options-answers-reasoning chains for the audio pair.

Audio translation 1 is [**audio translation 1**], audio description 1 is [**audio description 1**], video description 1 is [**video description 1**], audio translation 2 is [**audio translation 2**], audio description 2 is [**audio description 2**], video description 2 is [**video description 2**].

The questions must meet **the following criteria**:

1. The question must relate to audio pairs and cannot include any information or terms related to the video.

2. The category of the question should belong to one of the following categories, and the category must be specified: continuous scene reasoning task, multi-character interaction reasoning task, plot development reasoning task, anomaly detection and security task, time reasoning task.

3. The question should not have an answer that can be derived solely from the text title; it should require reasoning that combines the question and the audio.

4. The question should involve multi-step reasoning and be challenging, with options of moderate length for the answers.

5. The multiple-choice answers can independently reason for each audio's description but must include at least one reasoning question that is related to both audio segments simultaneously.

6. Your provided answers must conform to the following format: Question:..... Question Category: ...... Question Options: ...... Question Answer: ...... Question Reasoning Chain: ......

**Assistant:**
{response}

---

**Prompt of Data generation in MDAR-open**

**System:**
You are a helpful AI assistant, designed to provide useful answer to customers. The user will provide you with a video, audio, audio translation, audio description, and video description. Please refer to this desired output and provide your own unique response.

**User:** Propose an open-ended question for audio based on the given video, audio, audio translation, audio description, and video description. The requirement is an open-ended reference answer thought chain pair, where the audio translation is [**audio translation**], the audio description is [**audio description**], and the video description is [**video description**].

At the same time, the questions raised must meet **the following requirements**:

1. The requirements are only open-ended reasoning questions related to audio, and the video and provided text are only for reference in question setting and cannot be used for question setting (already questions, answers) and thought chain analysis. The questions can only be answered by understanding and reasoning about the audio.

2. The category of the question belongs to one of the following categories, and indicate which task it belongs to: time reasoning task, scene understanding task, character relationship and social reasoning task, event reasoning task, anomaly detection and security task.

3. The answer to the question cannot be found solely from the text question, it needs to be combined with the question and audio for reasoning.

4. The answer to an open-ended question is not unique, just provide a reference answer. The answer you provide should meet the following format: Question: .... Question Category: .... Reference answer: ....

**Assistant:**
{response}

---

**Prompt of Interference option generation**

**System:**
You are a helpful AI assistant, designed to provide useful answer to customers. The user will provide you with a video, audio, audio translation, audio description, and video description. Please refer to this desired output and provide your own unique response.

**User:** Based on the given QA pair (question and correct answer) and audio, video descriptions, and audio descriptions, generate 3 high-quality distractor options.

the audio translation is [**audio translation**], the audio description is [**audio description**], and the video description is [**video description**].

The distractor options should possess **deception** (appearing reasonable from a surface logic or common sense perspective, easily causing those who lack key information to misselect), **relevance** (closely adhering to the question theme without deviating from the core scene/-field), and **incorrectness** (clearly contradicting the correct answer or contradicting the facts in the background information), ultimately forming a complete set of multiple-choice options consisting of '1 correct answer and 3 distractor options'.

**Assistant:**
{response}

---

# E  MANUAL SCREENING CRITERIA

During the process of manual review, we set strict screening standards to ensure the high quality and consistency of the data.

**Question Screening Criteria**

- The grammar is correct, the wording is accurate, and there are no spelling errors. Ambiguous or easily misunderstood words should be avoided. The question must be logical in its expression, with all parts reasonably connected.

- The question must closely adhere to the theme and task scope set by the benchmark.

- The question should conform to the actual user questioning habits and context of the specific application scenario.

- Strictly check whether the question contains words such as `"video"` or `"description"`.

- Delete or modify QA pairs that violate the specified requirements to ensure that the questions are concise and clear.

**Answer Screening Criteria**

- The answer must correspond to objective facts, and for questions with clear factual basis, the answer must be accurate.

- The logical reasoning process of the answer must be correct, without any contradictions.

- For open-ended questions, the answer should provide sufficiently rich content. Check whether the answer contains words such as "video" or "description," as these are not allowed.

- The structure of the answer should be reasonable, with clear point-by-point responses and a complete structure.

**Overall QA Pair Screening Criteria**

- Consistency between the question and the answer; the answer must be a direct response to the question and not be off-topic.

- If there are multiple related QA pairs, these QA pairs must maintain consistency in content and logic.

- The quality of QA pairs should be relatively stable, avoiding situations where some QA pairs are of high quality while others are not. Check whether the answer contains words such as "video" or "description," as these are not allowed.

- Conduct random sampling or comprehensive checks of QA pairs to ensure that each pair meets the screening criteria.

# F  EVALUATION

## F.1  METRIC OF EVALUATION

**Exact Match:**

$$EM = \frac{1}{N} \sum_{i=1}^{N} 1 \ (if \ P_i = T_i) \tag{1}$$

**Jaccard Index:**

$$JI = \frac{1}{N} \sum_{i=1}^{N} \frac{|P_i \bigcap T_i|}{|P_i \bigcup T_i|} \tag{2}$$

**Mean Average Precision:**

$$Precision = \frac{1}{N} \sum_{i=1}^{N} \frac{|P_i \bigcap T_i|}{|P_i|} \tag{3}$$

**Mean Average Recall :**

$$Recall = \frac{1}{N} \sum_{i=1}^{N} \frac{|P_i \bigcap T_i|}{|T_i|} \tag{4}$$

N is the total number of samples. $P_i$ is the set of predicted labels for the i-th sample. $T_i$ is the set of ground truth labels for the i-th sample.

## F.2 PROMPT OF EVALUATION

---

**Prompt template for open-ended question answer scoring.**

**System:**
"You are a helpful and precise assistant for checking the quality of the answer."
"[Detailed Audio Description][Audio]"
"[Question][Question]"
"[The Start of Assistant 1s Answer][Assistant1]"
"[The End of Assistant 1s Answer]"
"[The Start of Assistant 2s Answer][Assistant2]"
"[The End of Assistant 2s Answer]"
"[System]"
"We would like to request your feedback on the performance of two AI assistants in response to the user question "
"and audio description displayed above. AI assistants are provided with detailed audio descriptions and questions."
"Please rate the helpfulness, relevance, accuracy, and comprehensiveness of their responses."
"Each assistant receives an overall score on a scale of 1 to 10, where a higher score indicates better overall performance. "
"Please output a single line containing only two values indicating the scores for Assistant 1 and 2, respectively. "
"The two scores are separated by a space."

**Assistant:**
{response}

---

**Detailed explanation of the grading criteria. refer to AIR-Bench (Yang et al., 2024)**

**System:**
"helpfulness": "Whether the response is helpful to the user and whether it can solve the user's problem"
"relevance": "Whether the response is related to the question and audio content"
"accuracy": "The accuracy of the response, whether it is factual"
"comprehensiveness": "The comprehensiveness of the response, whether it covers all aspects of the question"
"score range": "1-10 points, the higher the score, the better the performance"
"output format": "Output a line containing two scores, separated by a space"
**Assistant:**
{response}

---

## G DETAILED INFORMATION OF THE EVALUATED MODELS

In this section, we provide a detailed description of the models we selected, the models we did not select, and the implementation details, so as to ensure reproducibility.

### G.1 EVALUATED MODELS

- Qwen2-Audio-Instruct (Chu et al., 2024) is a new series of Qwen large audio-language models. It can accept inputs of various audio signals and perform audio analysis based on voice instructions or generate direct text responses. In this paper, we use the 7B Instruct model.

- Qwen-Audio-Chat (Chu et al., 2023) is a Large Audio Language Model developed by Alibaba Cloud. It can take multiple types of audio (including human speech, natural sounds, music, and singing voices) and text as inputs, and generate text as output, supporting a multi-task learning framework for various complex audio tasks.

- Audio Flamingo 2 (Ghosh et al., 2025) possesses advanced audio understanding and reasoning capabilities. In particular, it has professional audio reasoning ability and can understand long audio clips up to 5 minutes in duration.

- Audio Flamingo 3 (Goel et al., 2025) (AF3) is a fully open-source, state-of-the-art Large Audio Language Model (LALM), which consists of an AF-Whisper unified audio encoder, an MLP-based audio adapter, a decoder-only LLM backbone (Qwen2.5-7B), and a streaming TTS module (AF3-Chat). Audio Flamingo 3 can accept audio inputs with a duration of up to 10 minutes.

- Audio Flamingo 3 Chat. As the chat version of Audio Flamingo 3, it is capable of voice chat and multi-audio dialogue.

- Kimi-Audio-Instruct (KimiTeam et al., 2025) is designed as a general-purpose audio foundation model, which can handle a wide range of audio processing tasks within a single unified framework. It adopts mixed audio inputs (continuous acoustic signals + discrete semantic tokens) and an LLM core with parallel heads for text and audio token generation.

- Omni-R1 (Zhong et al., 2025) is an all-modal model that addresses the resolution issue through a dual-system architecture. Meanwhile, it proposes an end-to-end RL framework—Omni-R1 is built on Group Relative Policy Optimization (GRPO). The results demonstrate the first successful application of RL in large-scale all-modal reasoning and highlight a scalable path toward general-purpose foundation models.

- R1-AQA (Li et al., 2025) is an Audio Question Answering (AQA) model optimized through reinforcement learning using the Group Relative Policy Optimization (GRPO) algorithm.

- SALAMONN (Tang et al., 2024) is a Large Language Model (LLM) that supports speech, audio event, and music inputs, developed by the Department of Electronic Engineering of Tsinghua University and ByteDance. It can perceive and understand various types of audio inputs. In this paper, we use the 7B version.

- Audio-Reasoner (Xie et al., 2025) is an open-source project developed by a team from Tsinghua University, focusing on building a large language model that supports in-depth audio reasoning. Based on Qwen2-Audio-Instruct, this model incorporates structured chain-of-thought technology to achieve complex reasoning and multi-modal understanding of audio content.

- DeSTA2.5-Audio (Lu et al., 2025) is a general-purpose Large Audio Language Model (LALM) designed for robust auditory perception and instruction-following capabilities, without the need for task-specific audio instruction tuning.

- MiDashengLM (Dinkel et al., 2025) integrates the powerful Dasheng audio encoder with the Qwen2.5-Omni-7B Thinker decoder through a unique caption-based alignment strategy.

- GPT-4o Audio is a multi-modal speech interaction model launched by OpenAI. It not only supports mixed input and output of text and audio but also achieves multiple technological breakthroughs in emotion recognition, real-time response, speech synthesis, and other aspects. It is a representative closed-source speech model in the current first-tier category.

Table 8: Detailed results for each category of MDAR-multi. The best results are highlighted in **bold** and the second-best is underlined.

| Model | Size | Type | EM(%) | | | JI(%) | | |
|---|---|---|---|---|---|---|---|---|
| | | | PDR | CSR | MCIR | PDR | CSR | MCIR |
| Qwen2-Audio-Instruct | 7B | LALM | 2.08 | 4.42 | 0.00 | 26.48 | 22.33 | 12.93 |
| Qwen-Audio-Chat | 8.4B | LALM | 7.29 | 6.28 | 0.00 | 48.06 | 42.23 | 38.45 |
| Qwen2.5-Omni | 3B | OLMs | 18.75 | 26.96 | 17.24 | 50.78 | 59.31 | 49.71 |
| Qwen2.5-Omni | 7B | OLMs | 25.00 | 27.10 | 27.59 | 54.50 | 58.80 | 49.43 |
| GPT-4o Audio | - | LALM | **52.08** | **51.36** | **62.07** | **72.40** | **74.16** | **80.17** |
| GPT-4o mini Audio | - | LALM | 18.75 | 21.11 | 39.29 | 49.05 | 53.22 | 67.86 |
| DeSTA2.5-Audio | 8B | LALM | 14.58 | 15.69 | 10.34 | 41.41 | 45.45 | 43.39 |
| Kimi-Audio-Instruct | 7B | LALM | 12.50 | 16.26 | 13.79 | 47.92 | 45.67 | 47.70 |
| MiDashengLM | 7B | LALM | 11.46 | 8.42 | 6.90 | 42.92 | 38.83 | 27.01 |
| Omni-R1 | 7B | OLMs | 6.25 | 3.28 | 3.45 | 46.49 | 46.21 | 43.97 |
| R1-AQA | 7B | LALM | 1.04 | 2.71 | 0.00 | 12.50 | 15.50 | 8.91 |

| Model | Size | Type | Precision(%) | | | Recall(%) | | |
|---|---|---|---|---|---|---|---|---|
| | | | PDR | CSR | MCIR | PDR | CSR | MCIR |
| Qwen2-Audio-Instruct | 7B | LALM | 32.48 | 28.85 | 22.41 | 34.90 | 28.22 | 16.67 |
| Qwen-Audio-Chat | 8.4B | LALM | 53.00 | 47.45 | 44.20 | 64.84 | 56.67 | 54.02 |
| Qwen2.5-Omni | 3B | OLMs | 73.09 | 80.35 | 72.70 | 54.34 | 62.80 | 52.87 |
| Qwen2.5-Omni | 7B | OLMs | 73.51 | 81.45 | 67.82 | 57.38 | 61.85 | 50.00 |
| GPT-4o Audio | - | LALM | **78.65** | **84.44** | 85.92 | **77.26** | **77.72** | **83.91** |
| GPT-4o mini Audio | - | LALM | 67.01 | 73.18 | **88.10** | 51.82 | 55.31 | 70.24 |
| DeSTA2.5-Audio | 8B | LALM | 55.47 | 62.59 | 63.79 | 47.05 | 51.13 | 48.85 |
| Kimi-Audio-Instruct | 7B | LALM | 69.97 | 68.12 | 65.80 | 53.47 | 49.26 | 53.45 |
| MiDashengLM | 7B | LALM | 51.60 | 49.32 | 42.82 | 52.60 | 49.04 | 35.06 |
| Omni-R1 | 7B | OLMs | 51.20 | 51.17 | 45.98 | 62.50 | 63.85 | 60.92 |
| R1-AQA | 7B | LALM | 18.58 | 22.42 | 11.78 | 14.41 | 18.77 | 13.22 |

- Qwen2.5-Omni (Xu et al., 2025a) is an end-to-end multi-modal model designed to perceive multiple modalities, including text, images, audio, and video, while generating text and natural speech responses in a streaming manner.

All models are used in inference mode only, with code and settings consistent with the official inference code, and a temperature value of 0 , max length is 256.

### G.2    UNSELECTED MODELS

- Mellow (Deshmukh et al., 2025) is a small audio language model specifically designed for reasoning. In our experiments, this model does not possess Chinese language capabilities.
- GAMA (Ghosh et al., 2024a) is a large audio language model that combines advanced audio understanding and complex reasoning capabilities. Its core technical highlight lies in its unique model architecture and data processing method. In our experiments, this model does not possess Chinese QA (Question Answering) capabilities.

## H    DETAILED RESULTS OF MDAR-MULTI

In Table 8, the detailed results of each subtask of MDAR-multi are clearly visible, which undoubtedly provide highly convincing evidence for the argument regarding the difficulty of the MDAR benchmark and the difficulty of each subtask.

## I    OPTION DISTRIBUTION AND INSTRUCTION BIAS

In multiple-choice tasks, the distribution of options and the phrasing of instructions often have a significant impact on the model's selection behavior. If the distribution of option quantities is uneven, or if certain options appear significantly more frequently in the training data, the model may tend to select these high-frequency options, leading to an overestimation or bias in performance. Additionally, the way instructions are phrased can also significantly affect the model's reasoning results.

Table 9: Average accuracy of multiple-choice questions using four models with different prompts

| | Prompt | Qwen2-Audio -Instruct | Omni-R1 | Qwen2.5-Omni | R1-AQA |
|---|---|---|---|---|---|
| v1 | $choices_{str} = "\backslash\mathbf{n}"$ $.join(item["chosens"])$ $item["question"]+$ 选项如下: $+choices_{str}+$ 选择正确的选项。 | 17.93 | 37.29 | 36.13 | 26.13 |
| v2 | $choices_{str} = "\backslash\mathbf{n}"$ $.join(item["chosens"])$ $item["question"]+$ 选项如下: $+choices_{str}+$ 请直接给出正确选项的内容。 | 24.13 | 43.83 | 76.27 | 29.00 |
| v3 | $choices_{str} = "\backslash\mathbf{n}"$ $.join(item["chosens"])$ $item["question"]+$ 请从下列选项中选择正确的选项。 $+choices_{str}$ | 14.33 | 21.01 | 26.00 | 23.07 |
| v4 | $choices_{str} = "\backslash\mathbf{n}"$ $.join(item["chosens"])$ $item["question"]+$ 请从下列选择中直接给出正确选项的内容: $+choices_{str}$ | 22.00 | 18.55 | 73.67 | 24.93 |

To avoid these biases, we ensure balanced option distribution, mutually exclusive option content, and clear semantics in the benchmark design, and when necessary, we randomize the order of options to reduce the influence of instruction prompts on model outputs. In terms of instruction selection, we tested four different instruction prompts, and the results are shown in the table 9. Ultimately, we adopted the instruction that yielded the best results, ensuring the correct outputs and reviews for the vast majority of models.

## J CASE STUDY

In order to better analyze the model's performance on different tasks, we conducted a qualitative study of the model's prediction results. This section uses specific examples to demonstrate common types of failures, differences in task performance, and sources of errors, helping us understand the model's limitations and areas for improvement in real-world scenarios.

### J.1 TASKS CASES

To explore the challenges posed by different task types on the model, we further present cases in a multi-task scenario (cases in Table 10 and Table 11). The results indicate that the model performs well on structured question answering and simple classification tasks, but its performance significantly declines in tasks that require multi-step reasoning, cross-modal integration, or in real-world data containing noise. This suggests that the current model's generalization ability is still constrained by the distribution of the training data, posing challenges to cross-task transfer capabilities.

### J.2 ERRORS CASES

Table 12 summarizes the distribution of error types, including perception errors, reasoning errors, knowledge loss, and formatting errors. Our analysis shows that the highest proportion of inference errors is mainly manifested in the breakage of the multi-step reasoning chain, errors in causal judgment and insufficient integration of contextual information. Perceptual errors often occur in the case of noise or multiple speaker interference, and factual errors caused by lack of knowledge cannot be ignored, while format errors reflect the weakness of the model in output constraints and normalization.

Table 10: Case of each tasks in MDAR-main

| Tasks | Question and Option | Answer |
|---|---|---|
| **Scene Understanding** | 问题：综合分析音频中的各种声音元素，该场景最有可能发生在哪里？选项：["一个播放着欧美流行乐的西式快餐厅","一个提供自助餐并播放传统音乐的员工餐厅","一个正在举办婚礼、人声鼎沸的中式酒楼","一个安静的、仅提供素食的私人会所"], | "一个提供自助餐并播放传统音乐的员工餐厅" |
| **Social Relationships and Social Reasoning** | 问题：综合音频，以下哪种场景最贴切地描述了对话双方的身份和所处情境？选项：["一位导演正在指导演员，并对她的表演提出反馈","两位专业的配音演员在录音棚里对稿，背景音乐是配乐样本","一位表演系学生正在家中练习台词，她的朋友或伴侣在一旁听着","一对情侣在一家有现场钢琴演奏的高档餐厅里进行深刻的情感交流"] | "一位表演系学生正在家中练习台词，她的朋友或伴侣在一旁听着" |
| **Event Reasoning** | 问题：结合音频推断对话发生的直接原因最可能是什么？选项：["一名男性因工作压力过大而临时约朋友倾诉","一名男性因家庭矛盾冲动离家后偶遇老友","一名男性因目睹交通事故后情绪崩溃寻求安慰","一名男性因回忆触发痛苦往事而躲进童年避难所"] | "一名男性因回忆触发痛苦往事而躲进童年避难所" |
| **Temporal Reasoning** | 问题：根据音频推断出此次抢救大约是从何时开始的？选项：["晚上20点45分","晚上21点08分","晚上21点30分","晚上20点55分"] | "晚上21点08分" |
| **Anomaly Detection and Safety** | 问题：结合音频可以推断出当前任务最显著的特点是什么？选项：["任务已成功完成，正在进行事后汇报","任务规划存在重大分歧，团队正在激烈争论","这是一个演习场景，电子音是模拟结束的信号","任务具有极高的时间敏感性和风险，可能正处于倒计时阶段"] | "任务具有极高的时间敏感性和风险，可能正处于倒计时阶段" |

Table 11: Cases of each tasks in MDAR-multi

| Tasks | Question and Option | Answer |
|---|---|---|
| **Plot Development Reasoning** | 问题："综合两段音频中的对话和情境，可以对人物关系和核心情节做出哪些合理的推断？"选项：["在第一段音频中，女性角色指控男性角色的动机是，万文芳发现了该男性并非其亲生儿子万思臣"，"两段音频中的核心冲突是围绕一份亲子鉴定展开的，该鉴定是为了证明男性角色是女性角色的亲生父亲，以此来承担责任"，"第二段音频中提到的"亲子鉴定"很可能是对第一段音频中"你不是万思臣"这一指控的决定性验证，这份结果将证实男性的真实身份"，"从第一段音频中女性的质问"这不是你的复仇计划吗"可以推断，她承认了谋杀案是自己策划的，目的是为了向男性复仇"] | ["在第一段音频中，女性角色指控男性角色的动机是，万文芳发现了该男性并非其亲生儿子万思臣，因此他杀人灭口"，"第二段音频中提到的"亲子鉴定"很可能是对第一段音频中"你不是万思臣"这一指控的决定性验证，这份结果将证实男性的真实身份"] |
| **Continuous Scene Reasoning** | 问题：结合两段音频，关于这场冲突的规模和发展，可以得出哪些合理的推断？选项：["音频1中，沉重的撞击声和持续的坍塌声暗示了冲突可能发生在一座大型建筑或城墙附近，且该结构正在遭受猛烈攻击"，"音频2中的密集爆炸声和混乱的金属碰撞声，相比于音频1，表明冲突进入了更白热化、更混乱的近距离交战或轰炸阶段"，"综合两段音频，战斗的规模宏大，不仅限于小队交火，而是涉及了大规模杀伤性力量的全面战争"，"从音频1到音频2，冲突的强度明显减弱，声音从大规模破坏转为零星的个人战斗，表明战斗已接近尾声"] | ["音频1中，沉重的撞击声和持续的坍塌声暗示了冲突可能发生在一座大型建筑或城墙附近，且该结构正在遭受猛烈攻击"，"音频2中的密集爆炸声和混乱的金属碰撞声，相比于音频1，表明冲突进入了更白热化、更混乱的近距离交战或轰炸阶段"，"综合两段音频，战斗的规模宏大，不仅限于小队交火，而是涉及了大规模杀伤性力量的全面战争"] |
| **Multi-Character Interaction Reasoning** | 问题：综合两段音频，以下关于两位说话者互动动态的推断，哪些是合理的？选项：["在第一段音频中，年长者提及泡澡的回忆，其语气透露出他希望通过共同的过去来缓和当前略显紧张和尴尬的气氛"，"在第二段音频中，年长者讲述了更具体、甚至有些尴尬的童年趣事，这表明他可能在试探年轻人的真实反应，而不仅仅是单纯地怀旧"，"综合两段音频，年轻人从始至终以平淡、疏远的语气回应"不记得"，这种一致性暗示他的"遗忘"可能并非简单的记性不好，而是一种刻意的情感疏离或心理防御"，"两段音频整体氛围轻松愉快，对话显示两人正在愉快地分享童年糗事，关系非常融洽"] | ["在第一段音频中，年长者提及泡澡的回忆，其语气透露出他希望通过共同的过去来缓和当前略显紧张和尴尬的气氛"，"在第二段音频中，年长者讲述了更具体、甚至有些尴尬的童年趣事，这表明他可能在试探年轻人的真实反应，而不仅仅是单纯地怀旧"，"综合两段音频，年轻人从始至终以平淡、疏远的语气回应"不记得"，这种一致性暗示他的"遗忘"可能并非简单的记性不好，而是一种刻意的情感疏离或心理防御"] |

Table 12: Cases of each errors

| Errors | Question and Option | Answer | Output |
|--------|---------------------|--------|--------|
| **Perception Errors** | 问题：根据音频，对话最可能发生在以下哪个场景？选项：["喧闹的警察局大厅","作为临时据点的室内空间","车辆疾驰的马路边","人声鼎沸的露天夜市"], | "作为临时据点的室内空间" | "人声鼎沸的露天夜市" |
| **Reasoning Errors** | 问题：根据音频对话，可以推断出两位对话者的身份关系最可能是什么？选项：["图书管理员与一位正在抱怨的读者","一起处理新到货物的同事","正在进行安全演练的师生","送货员与负责签收的职员"] | "一起处理新到货物的同事" | 根据音频对话中的内容，可以听到对话涉及"新到的书"，同时还提到了"分类"这样的词汇。这些关键词很可能与图书馆的场景相关。因此，两位对话者的关系最可能是"图书管理员与一位正在抱怨的读者"。 |
| **Knowledge Gaps** | 问题：这段欢快的音乐最有可能出现在以下哪个节日的庆祝活动中？选项：["日本的盂兰盆节","巴西的狂欢节","美国的阿巴拉契亚民俗音乐节","西班牙的弗拉明戈艺术节"] | "美国的阿巴拉契亚民俗音乐节" | "巴西的狂欢节" |
| **Formatting Errors** | 问题：综合音频，最有可能是什么事件刚刚发生，从而引发了第一位说话者的这段言论？选项：["他刚刚阅读完一份冗长乏味的历史文献","有人刚刚完成了一次迅猛如风、精准无比的箭术或剑术展示","一位异国使者刚刚带着和平条约抵达","太阳刚刚升起，照亮了一片壮丽的风景"] | "有人刚刚完成了一次迅猛如风、精准无比的箭术或剑术展示" | "有人刚刚完成了——次迅猛如风，精准无比的箭术或剑术展示" |