# OpenReview forum: "MDAR: A Multi-scene Dynamic Audio Reasoning Benchmark"
_ICLR.cc/2026/Conference — ICLR 2026 Conference Withdrawn Submission_

### Official Review · Reviewer_zBaU · 2025-10-31

**Soundness:** 3
**Presentation:** 3
**Contribution:** 2
**Rating:** 2
**Confidence:** 4

**Summary:**

The paper proposes a benchmark, MDAR, to evaluate audio reasoning capabilities of LALMs.
MDAR emphasizes complex tasks with multi-scene and potentially dynamically evolving contexts.
MDAR contains 3K curated samples for 5 categories and 3 question types.
The paper evaluates MDAR on 26 SOTA LALMs and find these models have limitations on these tasks.

**Strengths:**

- Overall, the paper aims at studying a very important problem in the audio understanding domain. It remains a challenge to evaluate and enhance the complex reasoning abilities of current LALMs, and this paper proposes a solid benchmakr for more holistic evaluation of this area.

- The test data construction pipeline is very clear, transparent, and reproduce-able. The taxonomy of different types of test samples (different categories and question types) could benefit the community to conduct more fine-grained analysis and evaluation of their models.  These also complement existing benchmarks (e.g. MMAU and MMAR) that have different taxonomies (e.g. mainly on audio domains).

- The baseline evaluation is quite extensive. The paper evaluates many open and closed models and the results are valuable. In addition to just overall accuracies, the paper further conducts detail analysis such as error types in Fig 6. The results are very valuable as it tells us that most mistakes are related to mis-reasoning and some are mis-perception. These could benefit further RL design choices of LALM post training.

**Weaknesses:**

- As stated in L215, test samples are all from Chinese films and therefore the distribution can be biased. This could limit the versatility of the proposed benchmark, and may also be unfair to LALMs that are only trained on English corpus / data.

- While Fig 3 shows the distribution and other strengths of the proposed MDAR, it is hard to quantitively prove how much better MDAR is compared to existing benchmarks like MMAU and MMAR. To me it sounds more like a complement to those benchmarks. I suggest the authors to conduct a rigorous human study on how MDAR is better than prior benchmarks, despite that MDAR covers more skills and contains longer audio. Similarly, the paper claims the samples are complex, multi-scene, and the context can be dynamic, but these require rigorous proof.

- The evaluation of open-ended samples is based on LLM-as-a-judge. This can be unstable and unreliable in terms of rigorous comparison.

- The experimental analysis is not extensive at all. While there are main accuracies and error analysis, there is no in-depth analysis on different types of errors. For instance, why do the models have these errors? What are the causes of these errors -- capacity issues, hallucination, randomness when responding, or others? It worths studying these deeper information so that the community could further improve their models.

- Following the above point, the paper does not indicate readers how to build better LALMs. Should we use better SFT or better RL? Should we scale the base LLM or not? What types of data to curate to fill the gap? Does Fig 6 indicate we should blend more text-specific data? How to design efficient methods for reliable understanding of longer audio? I would expect the paper to show some route for building better LALMs -- while the ultimate goal is always to have more and higher-quality data in the end -- the benchmark analysis should guide us to reach the goal more efficiently and effectively with principled recipes and experiences.

**Questions:**

See weaknesses.

---

> ### Author Response · Authors · 2025-11-20
> **Response to reviewer zBaU(1/2）**
>
> We are extremely grateful for your valuable suggestions. We have addressed each of the issues you raised and incorporated the corresponding revisions into our latest manuscript.
>
> Weakness 1
> > As stated in L215, test samples are all from Chinese films and therefore the distribution can be biased. This could limit the versatility of the proposed benchmark, and may also be unfair to LALMs that are only trained on English corpus / data..
>
> **Our Response:**
> Thank you very much for your valuable feedback! What we need to clarify is that regarding the question of fairness in the evaluation, we have already considered the possibility of conducting Chinese question answering and Chinese proficiency in the process of selecting the model for evaluation, as shown in Appendix (G.2). Therefore, the models participating in the evaluation in the article do not have this issue. We also look forward to the emergence of more powerful models trained using Chinese corpus in the speech community in the future.
>
> Weakness 2
> > While Fig 3 shows the distribution and other strengths of the proposed MDAR, it is hard to quantitively prove how much better MDAR is compared to existing benchmarks like MMAU and MMAR. To me it sounds more like a complement to those benchmarks. I suggest the authors to conduct a rigorous human study on how MDAR is better than prior benchmarks, despite that MDAR covers more skills and contains longer audio. Similarly, the paper claims the samples are complex, multi-scene, and the context can be dynamic, but these require rigorous proof.
>
> **Our Response:**
> Thank you for your valuable feedback!
> What we want to clarify is that our benchmark focuses on evaluating dynamic scenario inference, including continuous scenarios, causal scenarios, and multi scenario tasks. We believe that this is a key capability requirement for AI agents in real scenarios, which is not emphasized by current benchmarks. MMAU, MMAR, and MMAU-Pro (partially the same as our goal) all focus on more diverse perceptual samples, including counting, classification, and other perceptual tasks. Our evaluation dimension focuses on complex logical inference in dynamic scenarios, and is further divided into three tasks for evaluation. Our benchmark also has longer speech, which means it contains more information. The following table compares the relevant works.
>
> |   | Mixed audio  | multi-audio  | Open-ended  |  multi-scene within one audio | chinese  |  Instruct following |
> | ------------ | ------------ | ------------ | ------------ | ------------ | ------------ | ------------ |
> | AudioBench | × | × | √ | × | × | × |
> | AIR-Bench  | × | × | √ | × | × | × |
> | MMAU  | ×  | × | √ | × | × | × |
> | MMAR |  √  | × | × | × | ×  | ×  |
> | **MDAR（ours）** | **√** | **√** | **√** | **√** | **√** | **√** |
>
> Weakness 3
> > The evaluation of open-ended samples is based on LLM-as-a-judge. This can be unstable and unreliable in terms of rigorous comparison.
>
> **Our Response:**
> Based on your suggestion, we conducted the following supplementary experiments to evaluate the correlation between language models (LLM) and human assessment. We first randomly selected 10% of the samples and manually rated the responses of three models (Qwen2-Audio-7B, Qwen25-Omni-7B, and GPT4o-Audio). The rating range is 0 to 5 points, with 5 points indicating a completely correct answer and clear expression. We calculated the Pearson correlation coefficient between human evaluation and model evaluation, and the results showed that the Pearson correlation coefficients of all three models were greater than 0.9, indicating a high degree of correlation and credibility between human evaluation and model evaluation.In addition, we conducted pairwise paired sample t-tests among the three models for all samples of the open benchmark. The test results showed that the p-values of all model pairs were less than 0.05, indicating significant differences in the evaluation scores between different models, further demonstrating the effectiveness of our evaluation. The following table lists the specific values of these two experiments.
>
> |   | samples  |  Qwen2-Audio-7B | Qwen25-Omni-7B  | GPT4o-Audio  |
> | ------------ | ------------ | ------------ | ------------ | ------------ |
> | Pearson Correlation Coefficient  | 50  | 0.95312  | 0.9268  | 0.9714  |
> |   | **samples**  | **Qwen2-Audio-7B v.s. Qwen25-Omni-7B**  | **Qwen25-Omni-7B v.s. GPT4o-Auido**  | **GPT4o-Auido v.s. Qwen2-Audio-7B**  |
> | t-test  |  500 | T: 6.421, P: 0.000  |  T: -11.866, P: 0.000 | T: -8.389, P: 0.000  |

---

> > ### Author Response · Authors · 2025-11-20
> > **Response to reviewer zBaU(2/2）**
> >
> > Weakness 4
> > >  There is no in-depth analysis on different types of errors. For instance, why do the models have these errors? What are the causes of these errors -- capacity issues, hallucination, randomness when responding, or others? It worths studying these deeper information so that the community could further improve their models.
> >
> > **Our Response:**
> > Thank you for your valuable feedback!
> > We only provided the main accuracy and error analysis in the paper, and then we analyzed a large number of cases for further discussion. We divide the errors in the model into four types, with inference errors accounting for the majority according to statistics. The causes of each type of model error are different. For LALMs, randomness and instability are still a big problem. In the prompt experiment, we found that different prompts can lead to significant differences in accuracy, and the same prompt can also produce different answers in different runs. This is because LALMs are more inclined to answer the content (refer to MMAU). Comparing the text can answer A/B/C/D. The capacity issue will be discussed in the next response.
> >
> > The problem of hallucinations mainly focuses on fabricating non-existent sound events or even non-existent options (rarely, in some models). In addition, models such as SALMONN fail to hear the key content in many problems, resulting in selection errors, indicating insufficient granularity of perception. This may point out some directionality for model training - finer perceptual speech descriptions or finer grained QA problem training - different formats of prompts to enhance the stability of the model for different format questions, especially when focusing on text requirements beyond speech.
> >
> > Weakness 5
> > > The paper does not indicate readers how to build better LALMs. Should we use better SFT or better RL? Should we scale the base LLM or not? What types of data to curate to fill the gap? Does Fig 6 indicate we should blend more text-specific data? How to design efficient methods for reliable understanding of longer audio?
> >
> > **Our Response:**
> > Thank you for your valuable feedback!
> > We will provide some insights into the construction of LALMs based on the benchmark of this article. Firstly, we will classify and analyze the evaluated models from different perspectives.
> >
> > By dividing the model into RL based on the training process, we compared R1-AQA with speech data RL and Qwen2-Audio-7B without RL. We observed that R1-AQA performed better than qwen2Audio-7b with RL. In the original R1-AQA text, it was suggested that SFT improvement was limited, indicating that RL for speech data may be effective. In MDAR testing, the improvement brought by RL currently appears to be minimal. I speculate that the main reason is that LALMs require a more powerful base model for RL, and I believe that the effect of RL will be more significant at that time.
> >
> > At the same time, we also sorted out the training stage and parameter size of the model. The 7B model has improved performance compared to the 3B model under the same architecture, but as shown in the error analysis, the 7b model mainly analyzed in this article has the second largest proportion of perception errors. To solve this problem, instead of continuing to increase the number of parameters, I think we should focus more on the quality of the data (such as fine-grained issues) and a better post training process.
> >
> > In the cascade experiment, we observed a significant improvement in the accuracy of the inference model using deepseek-V3, which not only demonstrates the insufficient text inference ability of existing LALMs, but also provides us with some potentially important insights in terms of data. On the premise of not losing perceptual ability, when training LALMs, we should focus on data diversity (to alleviate the instability shown in prompt experiments), as well as text specific abilities. I believe that adding code or math training can help improve audio inference ability.
> >
> > At present, the Audio flamingo model we are testing can support long audio understanding, but its performance on the MDAR benchmark is not as good as other models. We believe that it sacrifices some short speech performance while maintaining long audio capability. How to maintain the performance balance between long and short audio is a very good exploration topic.
> >
> > Finally, we sincerely thank you again for all the valuable suggestions you have provided for our manuscript. All the analysis and arguments will be added to the new version of the manuscript. We believe that MDAR's open source work can promote the better development of the audio community. If you have any further opinions, please let us know and we will do our best to solve them.

---

### Official Review · Reviewer_JsvT · 2025-11-01

**Soundness:** 3
**Presentation:** 3
**Contribution:** 3
**Rating:** 6
**Confidence:** 5

**Summary:**

This paper introduces MDAR, a new benchmark designed to evaluate multi-audio, multi-scene reasoning abilities in LALMs. Unlike prior benchmarks that focus on single-audio or localized perception tasks, MDAR tests whether models can reason compositionally across multiple audio scenes involving speech, environmental sounds, and music. The benchmark aims to push beyond simple perception toward holistic scene understanding, including temporal order, causal interactions, and semantic relationships between events. The final dataset contains 8,524 QA pairs across five task types: causal reasoning, temporal reasoning, multi-speaker dialogues, cross-scene summarization, and auditory anomaly detection.

**Strengths:**

- To the best of my knowledge, this is the first benchmark to target multi-scene dynamic reasoning, a previously unaddressed challenge for LALMs. MMAU-Pro released recently addresses this, but I understand it may be parallel work.
- Well-structured taxonomy of reasoning skills.
- Balanced synthetic-real composition pipeline ensuring diversity and controlled difficulty.
- The dataset also has an open and MCQ part, which is exciting and important.
- Comprehensive evaluation across open- and closed-source systems.

**Weaknesses:**

- Heavy reliance on synthetic compositions may limit real-world generalization.
- I do not have many weaknesses to point out.
- Is the dataset in Mandarin? I see the benchmark and looks in Supplementary and looks like its all Mandarin?

**Questions:**

- I am curious how multi-audio evaluation was done on models like AF2 and AF3 and maybe Qwen-3-omni, I did not find a straightforward way to do this.

---

> ### Author Response · Authors · 2025-11-20
> **Response to reviewer JsvT**
>
> We are extremely grateful for your valuable suggestions.
>
> Weakness 1
> > Heavy reliance on synthetic compositions may limit real-world generalization.
>
> **Our Response:**
> Thank you very much for your valuable feedback! Our synthesis is only aimed at problem generation, and the design of problem categories is still subject to manual discussion. The selection of scenes is based on movies as a medium, which still reflects real scenes. Different themes of movie clips provide us with diversified real scenes as benchmarks.
>
> Weakness 2
> > Is the dataset in Mandarin? I see the benchmark and looks in Supplementary and looks like its all Mandarin?
>
> **Our Response:**
> Thank you for your question. The text questions we use for QA are in Mandarin, and our audio clips use movie soundtracks that not only include Mandarin, but also diverse accents and dialects of Chinese. We will present specific dialect data in the new version of the manuscript in the future.
>
> Weakness 3
> > I am curious how multi-audio evaluation was done on models like AF2 and AF3 and maybe Qwen-3-omni, I did not find a straightforward way to do this.
>
> **Our Response:**
> Thank you for your question. AF2 and AF3 are currently unable to perform multi audio evaluation, and our experiment did not involve multi audio evaluation of these two models. The multi audio evaluation of Qwen2.5-OMni is because Qwen2.5-OMni supports playing multiple audio in the conversion input.
>
>
> Finally, we sincerely appreciate all the valuable suggestions you have provided for our manuscript, as well as your recognition of our work. We believe that the open source work of MDAR can promote the better development of the audio community. If you have any further opinions, please let us know and we will do our best to solve them.

---

### Official Review · Reviewer_g5UY · 2025-11-02

**Soundness:** 2
**Presentation:** 3
**Contribution:** 2
**Rating:** 4
**Confidence:** 5

**Summary:**

This paper introduces MDAR (Multi-scene Dynamic Audio Reasoning), a benchmark targeting complex, real-world audio reasoning across three complementary test suites: MDAR-main (1,500 single-choice QAs across five categories), MDAR-open (500 open-ended QAs scored by an LLM judge), and MDAR-multi (825 multi-audio, multi-answer QAs emphasizing cross-clip reasoning). Data are drawn from ~20–40s segments of Chinese movies with speaker diarization and global clustering; FunASR supplies transcripts; Gemini-2.5-Pro and Qwen-2.5-VL generate multimodal descriptions and help author questions/distractors under an expert-designed taxonomy and multi-step QA screening. MDAR reports diverse metrics (regex-based accuracy for single-choice, LLM-judge 0–10 for open-ended, and EM/JI/Precision/Recall for multi-audio). Baselines show the task is hard: best non-cascaded model in MDAR-main (Qwen2.5-Omni) reaches ~76.7%; MDAR-open tops out near 7.46/10; ablations (e.g., replacing audio with Gaussian noise) substantially degrade accuracy, underscoring audio dependence.

**Strengths:**

- The three suites (single-choice, open-ended, multi-audio) cover perception --> understanding --> cross-clip reasoning, with five high-level categories (scene, social, event, temporal, safety).
- Well-defined metrics: Regex-based accuracy for MDAR-main, an LLM-judge rubric for MDAR-open, and EM/JI/Precision/Recall for MDAR-multi are specified.
- The data generation pipeline is strong and well thought out.

**Weaknesses:**

- One of my biggest concerns is that all the QA are in Chinese in the supplementary material, but the examples show in the paper are in English. If the questions are fed to models like Audio Flamingo 3, which are not trained on QA pairs in the specific language, that might not be a fair evaluation. Also, the authors should explicitly mention that the QAs are in Chinese.
- The evaluated methods score significantly higher than they do on the benchmarks already released, which questions the difficulty of the benchmark.
- Since, most of the benchmarks is derived from movie clips, it would benefit from having questions around music (background/foreground).
- No correlation is shown between the Human & LLM as a judge. What is the correlation (e.g., Spearman/Pearson, Krippendorff’s α) between the LLM-judge scores and a human panel across categories?
- Mixing accuracy (closed-set), judge scores (open-ended), and set metrics (multi-audio) complicates aggregate interpretation; there’s no unified “overall” score or calibration across metrics.

**Questions:**

See weakness section.

---

> ### Author Response · Authors · 2025-11-20
> **Response to reviewer g5UY**
>
> We greatly appreciate the valuable suggestions you provided. We have addressed each issue you raised and incorporated the corresponding revisions into our latest manuscript.
>
> Weakness 1
> > One of my biggest concerns is that all the QA are in Chinese in the supplementary material, but the examples show in the paper are in English. If the questions are fed to models like AF3, which are not trained on QA pairs in the specific language, that might not be a fair evaluation. Also, the authors should explicitly mention that the QAs are in Chinese.
>
> **Our Response:**
> Thank you very much for your valuable feedback! We would like to clarify the following:
> 1. In the data processing section of our paper, we mentioned that we used Chinese movie clips for QA production. To avoid any further misunderstanding, we have reiterated this point in the revised version of the manuscript.
> 2. Regarding the concerns about evaluation fairness, we have already considered whether the models are capable of Chinese QA and have Chinese proficiency during the model evaluation process (see Appendix G.2). Therefore, the models involved in the evaluation in our paper do not have this issue.
>
> Weakness 2
> > The evaluated methods score significantly higher than they do on the benchmarks already released, which questions the difficulty of the benchmark.
>
> **Our Response:**
> Thank you very much for your valuable feedback! The good performance of SOTA is limited to the highest accuracy of single voice, but the single audio result table (tab. 1) reflects the discrimination of different LALMs. Most LALMs, including AF3, Qwen2-audio struct, have an accuracy around 20-40, which indicates that the dimensions we evaluate are difficult. Moreover, in multi voice and open benchmarks, this polarization phenomenon is more obvious. The closed source GPT4o-Audio (SOTA) is only 50%, and some LALMs are even as low as 10%, with extremely high discrimination. This is the key to measuring speech AI agents.
>
> Weakness 3
> > Since, most of the benchmarks is derived from movie clips, it would benefit from having questions around music (background/foreground).
>
> **Our Response:**
> Thank you very much for your valuable feedback! In the process of designing the problem, we considered a comprehensive range of task types and subjects to be examined, including sound, speech, and music. Questions about music were also included in the benchmark.
>
> Weakness 4
> > No correlation is shown between the Human & LLM as a judge. What is the correlation (e.g., Spearman/Pearson, Krippendorff’s α) between the LLM-judge scores and a human panel across categories?
>
> **Our Response:**
> Thank you very much for your valuable feedback! Based on your suggestion, we have conducted a correlation experiment between LLM and Human as follows. Firstly, we randomly selected 10% of samples and manually scored the responses of Qwen2-Audio-7B, Qwen25-Omni-7B, and GPT4o-Audio models. The range of manual scoring is 0-5 points, where 5 points means the answer is completely correct and the expression is clear. Then, we provided Pearson coefficients, all of which are greater than 0.9, indicating that our human evaluation is highly correlated and reliable with LLM evaluation. At the same time, we conducted paired sample t-tests between all three models based on an open benchmark, and the P-values were all less than 0.05, indicating significant differences in the evaluation scores of different models. The evaluation is completely effective, and the table below provides specific values for two experiments.
>
> |   | samples  |  Qwen2-Audio-7B | Qwen25-Omni-7B  | GPT4o-Audio  |
> | ------------ | ------------ | ------------ | ------------ | ------------ |
> | Pearson Correlation Coefficient  | 50  | 0.95312  | 0.9268  | 0.9714  |
> |   | **samples**  | **Qwen2-Audio-7B v.s. Qwen25-Omni-7B**  | **Qwen25-Omni-7B v.s. GPT4o-Auido**  | **GPT4o-Auido  v.s. Qwen2-Audio-7B**  |
> | t-test  |  500 | T: 6.421, P: 0.000  |  T: -11.866, P: 0.000 | T: -8.389, P: 0.000  |
>
> Weakness 5
> > Mixing accuracy (closed-set), judge scores (open-ended), and set metrics (multi-audio) complicates aggregate interpretation; there’s no unified “overall” score or calibration across metrics.
>
> **Our Response:**
> Thank you very much for your valuable feedback! This benchmark includes three different types of questions, and we should design different evaluation criteria and indicators. The focus of the three types of questions is different, and we cannot weight them solely based on a single accuracy score. We hope to have targeted and fair comparisons when using our benchmark.
>
> Finally, we sincerely appreciate all the valuable suggestions provided by the reviewers for our manuscript. We will ensure that all revisions are included in the next version of our manuscript. We believe that the open source work of MDAR can promote the better development of the audio community. If you have any further opinions, please let us know and we will do our best to solve them.

---

### Official Review · Reviewer_eGBs · 2025-11-10

**Soundness:** 3
**Presentation:** 3
**Contribution:** 2
**Rating:** 2
**Confidence:** 4

**Summary:**

The paper introduces MDAR which is a benchmark for multi-scene, dynamically evolving audio reasoning. It contains 3000 question-ans pairs spanning five different categories (Scene Understanding, Social Relationships, Event Reasoning, Temporal Reasoning, Anomaly and  Safety) and three formats: single choice on a single audio (also called MDAR-main), open-ended responses (MDAR-open), and (iii) a new multi-audio, multiple-choice setting (MDAR-multi). The data prep pipeline takes 500 Chinese films, samples 20 - 40s clips, applies speaker diarization and global segment clustering, then uses ASR plus LLMs (Gemini/Qwen) with expert screening to finalize questions and distractors.

**Strengths:**

- Dynamic, multi-scene coverage across five reasoning axes and three formats
- Longer and information dense clips (average 25.11s) compared to the prior work, raising temporal reasoning difficulty in the benchmakr
- Comprehensive evaluation of 26 models
- Systematic evaluation of both perceptual and high-level reasoning abilities
- Highlights key areas for improvement in next-generation audio reasoning agents

**Weaknesses:**

- Chinese movies as the metadata source biases the benchmark towards a particular accent/way of speaking. The benchmark should be diverse in the data to avoid biased results
- The SOTA performance on the benchmark closing in on 80%
- (General comment) The paper has a low excitement factor and raises a concern of lack of novelty with it being yet another benchmark. Even the paper/analysis/diagrams format exactly follows MMAU
- MMAU pro has not been included in the comparison with existing benchmarks
- error analysis has been done by Gemini-2.5-flash - ideally the errors should be identified and categorised by a human to get an accurate analysis. what prompt was used for this analysis? what are the examples of gemini-2.5-flash failing to do the required job?

**Questions:**

1. How do you define complex reasoning and how is it different from reasoning? The paper mentions the word complex multiple times but it needs to define what it means for audio
2. How difficult is it to improve performance on the benchmark?
3. fig 3b total questions don't sum up to 3000. can you confirm?
4. the paper selects best prompt for accuracy, how sensitive are rankings to different prompts?
5. how much would the human eval score be?

---

> ### Author Response · Authors · 2025-11-20
> **Response to reviewer eGBs(1/2)**
>
> We are extremely grateful for your valuable suggestions. We have addressed each of the issues you raised and incorporated the corresponding revisions into our latest manuscript.
>
> Weakness 1
> > Chinese movies as the metadata source biases the benchmark towards a particular accent/way of speaking. The benchmark should be diverse in the data to avoid biased results.
> > MMAU pro has not been included in the comparison with existing benchmarks.
>
> **Our Response:** Thank you for your valuable suggestions! We would like to clarify that there is currently a severe shortage of Chinese datasets in the speech field. This includes test benchmarks such as MMAU/MMAR, and even the newly released MMAU-PRO (a contemporary work), which are all in English QA. We hope that this benchmark can encourage researchers in the speech community to pay more attention to Chinese data (including dialects) when training LALM, and we also hope that the Chinese understanding and reasoning abilities of speech models will continue to improve.
>
> Weakness 2
> > The SOTA performance on the benchmark closing in on 80%
> > (General comment) The paper has a low excitement factor and raises a concern of lack of novelty with it being yet another benchmark. Even the paper/analysis/diagrams format exactly follows MMAU
> > How do you define complex reasoning and how is it different from reasoning? The paper mentions the word complex multiple times but it needs to define what it means for audio
> > How difficult is it to improve performance on the benchmark?
>
> **Our Response:**Thank you for your valuable suggestions! We would like to clarify the following:
> 1. Our benchmark focuses on evaluating reasoning in dynamic scenarios, including continuous scenarios, causal scenarios, and multi-scenario tasks. We believe this is a key capability requirement for AI agents in real-world settings, which is not the focus of current benchmarks. MMAU, MMAR, and MMAU-PRO (partially aligned with our goals) focus more on diverse perception samples, including counting, classification, and other perception tasks. In contrast, our evaluation emphasizes complex logical reasoning in dynamic scenarios, with the progression assessed through three types of tasks. By "complex," we mean that our benchmark includes three types of tasks: fully mixed audio, multi-audio comprehension, and open-ended question answering, examining multiple scenarios within a single audio file. Additionally, we assess the model's ability to follow instructions and its reasoning capability in Chinese. The table below compares relevant works.
> |   | Mixed audio  | multi-audio  | Open-ended  |  multi-scene within one audio | chinese  |  Instruct following |
> | ------------ | ------------ | ------------ | ------------ | ------------ | ------------ | ------------ |
> | AudioBench | × | × | √ | × | × | × |
> | AIR-Bench  | × | × | √ | × | × | × |
> | MMAU  | ×  | × | √ | × | × | × |
> | MMAR |  √  | × | × | × | ×  | ×  |
> | **MDAR（ours）** | **√** | **√** | **√** | **√** | **√** | **√** |
> 2. The reviewers pointed out that the SOTA performance comparison is only good in terms of the highest accuracy for single-language speech, but the single-audio results table (Table 1) shows the differentiation among different LALMs. Most LALMs, including AF3 and Qwen2-audio-instruct, have an accuracy around 20–40, which indicates that the dimension we are evaluating is challenging. Moreover, in multilingual and open benchmarks, this polarization phenomenon is even more pronounced. The closed-source GPT4o-Audio (SOTA) only achieves 50%, and some LALMs are even below 10%, showing a significant differentiation. This is precisely what we aim to measure in evaluating speech AI agents.
> 3. We sincerely appreciate your noting that our article’s style is similar to pioneering excellent work like MMAU. We will add references to MMAU-pro in our work. We hope readers will focus on the motivation behind our proposed benchmark and our goal to promote contributions to the open-source speech community.
>
> Weakness 3
> > error analysis has been done by Gemini-2.5-flash - ideally the errors should be identified and categorised by a human to get an accurate analysis. what prompt was used for this analysis? what are the examples of gemini-2.5-flash failing to do the required job?
>
> **Our Response:**Thank you for your valuable suggestions! Based on your suggestion, we have updated the prompt words for misclassification using Gemini-2.5-flash to the latest version of the manuscript.
> > 有一段给定的音频，对此提出的问题是{}，给出的选项是{}，正确的答案是{}，正确的思维链是{}，而A模型给出的模型输出是{}，请根据以下说明和给定的信息判断模型的回答不正确的错误类型，直接输出错误类型。说明:1、回答的错误类型分为感知错误、推理错误、知识缺失、格式错误 2、感知错误指模型对输入音频的原始内容或特征产生错误理解，比如把“爆炸声”听成“掌声”；推理错误指模型已正确感知原始信息，但在逻辑推导或因果/时序/多步推理中出现缺陷，已知“人物A先离开，后出现枪声”却得出“枪声导致A离开”，根因：逻辑链断裂、归纳/演绎错误、冲突消解失败；知识缺失指模型缺乏完成任务所需的先验或外部知识，导致无法给出或给出错误结论，根因：训练语料未覆盖、事实库缺失、常识盲区；格式错误指模型输出不满足任务要求的格式或语法规范，导致无法解析或评分，比如选对了正确答案但没有完全一样输出导致没办法识别为正确
> item["question"],item["chosens"],item["answer"],item["chain-of-thought"],item["model_output"]

---

> > ### Author Response · Authors · 2025-11-20
> > **Response to reviewer eGBs(2/2)**
> >
> > Weakness 4
> > > fig 3b total questions don't sum up to 3000. can you confirm?
> >
> > **Our Response:**
> > We apologize for providing incorrect data and the correct data has now been updated in the latest version of the manuscript.
> >
> > |   | total  | single  | multi  | open-end  |
> > | ------------ | ------------ | ------------ | ------------ | ------------ |
> > | number  |  3000 | 1500  | 1000  |  500 |
> >
> > Weakness 5
> > > the paper selects best prompt for accuracy, how sensitive are rankings to different prompts?
> >
> > **Our Response:**
> > Thank you for your valuable suggestions! Following your suggestions, we have added tables of monolingual results for four different prompts and updated them in the latest version of the PDF. The tables show the impact of different prompts on the results, and in this paper, we use the prompt with the highest accuracy and the least impact as the result prompt.
> >
> > |   | prompt | Qwen2-Audio-7B  | Omni-R1  |  Qwen25-Omni-7B | R1-AQA  |
> > | ------------ | ------------ | ------------ | ------------ | ------------ | ------------ |
> > | version1  | choices_str = "\n ".join(item["chosens"]) item["question"]+"选项如下:"+choices_str+"选择正确的选项。"  | 17.93%  |  37.29% |  36.13% | 26.13%  |
> > |  version2 | choices_str = "\n ".join(item["chosens"]) item["question"]+"选项如下:"+choices_str+"请直接给出正确选项的内容。"  |  24.13% | 43.83%  | 76.27%  | 29.00%  |
> > |  version3 | choices_str = "\n ".join(item["chosens"]) item["question"]+"请从下列选项中选择正确的选项。"+choices_str | 14.33%  | 21.01%  |  26.00% |  23.07% |
> > |  version4 | choices_str = "\n ".join(item["chosens"])  item["question"]+"请从下列选择中直接给出正确选项的内容："+choices_str |   22.00% |  18.55% |  73.67% | 24.93%  |
> >
> > Weakness 6
> > > how much would the human eval score be?
> >
> > **Our Response:**
> > Thank you for your valuable comments! Based on your suggestions, we conducted a manual evaluation. Due to time constraints, we will provide the full scores from the human experimental evaluation as a reference in subsequent updates. At the same time, regarding the stability of LLM evaluation and its correlation with human evaluation, we performed Pearson correlation and t-test experiments. The table below demonstrates that our human evaluation is highly correlated and reliable with LLM evaluation, and that the assessment is significantly valid.
> >
> > |   | samples  |  Qwen2-Audio-7B | Qwen25-Omni-7B  | GPT4o-Audio  |
> > | ------------ | ------------ | ------------ | ------------ | ------------ |
> > | Pearson Correlation Coefficient  | 50  | 0.95312  | 0.9268  | 0.9714  |
> > |   | **samples**  | **Qwen2-Audio-7B v.s. Qwen25-Omni-7B**  | **Qwen25-Omni-7B v.s. GPT4o-Auido**  | **GPT4o-Auido  v.s. Qwen2-Audio-7B**  |
> > | t-test  |  500 | T: 6.421, P: 0.000  |  T: -11.866, P: 0.000 | T: -8.389, P: 0.000  |
> >
> >
> > Finally, we sincerely thank the reviewers for all the valuable suggestions provided on our manuscript. We will ensure that all revisions are incorporated into the next version of our manuscript. We believe that MDAR's open-source work can help advance the audio community, and if you have any further comments, please let us know, and we will do our best to address them.

---

### Note · Authors · 2025-12-27

I have read and agree with the venue's withdrawal policy on behalf of myself and my co-authors.